# Decay and renormalization of a longitudinal mode in a quasi-two-dimensional antiferromagnet

Seung-Hwan Do [1,9✉], Hao Zhang [1,2,9], Travis J. Williams [3], Tao Hong [3], V. Ovidiu Garlea [3], J. A. Rodriguez-Rivera [4,5], Tae-Hwan Jang [6], Sang-Wook Cheong [6,7], Jae-Hoon Park [6,8], Cristian D. Batista [2] & Andrew D. Christianson [1]

An ongoing challenge in the study of quantum materials, is to reveal and explain collective quantum effects in spin systems where interactions between different modes types are important. Here we approach this problem through a combined experimental and theoretical study of interacting transverse and longitudinal modes in an easy-plane quantum magnet near a continuous quantum phase transition. Our inelastic neutron scattering measurements of $Ba_2FeSi_2O_7$ reveal the emergence, decay, and renormalization of a longitudinal mode throughout the Brillouin zone. The decay of the longitudinal mode is particularly pronounced at the zone center. To account for the many-body effects of the interacting low-energy modes in anisotropic magnets, we generalize the standard spin-wave theory. The measured mode decay and renormalization is reproduced by including all one-loop corrections. The theoretical framework developed here is broadly applicable to quantum magnets with more than one type of low energy mode.

[1] Materials Science and Technology Division, Oak Ridge National Laboratory, Oak Ridge, TN, USA. [2] Department of Physics and Astronomy, University of Tennessee, Knoxville, TN, USA. [3] Neutron Scattering Division, Oak Ridge National Laboratory, Oak Ridge, TN, USA. [4] Department of Materials Sciences, University of Maryland, College Park, MD, USA. [5] NIST Center for Neutron Research, Gaithersburg, MD, USA. [6] MPPHC-CPM, Max Planck POSTECH/Korea Research Initiative, Pohang, Republic of Korea. [7] Rutgers Center for Emergent Materials and Department of Physics and Astronomy, Rutgers University, Piscataway, NJ, USA. [8] Department of Physics, Pohang University of Science and Technology, Pohang, Republic of Korea. [9] These authors contributed equally: Seung-Hwan Do, Hao Zhang. ✉email: doh1@ornl.gov

One of the strongest signatures of collective quantum behavior is the spontaneous quasi-particle decay in interacting bosonic systems, as observed in superfluids[1–3] and quantum magnets[4–8]. In the latter case, spontaneous magnon decay has been studied in a growing number of lattice geometries and model systems where large quantum fluctuations enhance this many-body effect[9,10]. A key finding of these studies is that the strong decay process is accompanied by a significant renormalization of the overall spectrum[11–16]. This spectral renormalization leads to measurable effects in the thermal dynamic and transport properties[17], which are inexplicable without considering the renormalization of the quasi-particle mass. At the same time, the renormalization of the spectra opens an avenue to understand quantum systems since the renormalized single-magnon dispersion provides a stringent test for theories that attempt to describe magnon decay. In other words, approaches that do not fully incorporate these many-body effects will not yield correct values of the interaction parameters extracted from experimental studies.

An important question is how to understand quasi-particle decay in quantum magnets when there is more than one type of low-energy mode, i.e. when the parent particles are not of the same type as the daughter particles. Anisotropic magnets with spin $S \geq 1$ provide a common example of this situation. The additional fluctuations (quadrupolar for $S \geq 1$, octupolar for $S \geq 3/2$, etc.) can generate modes which are not captured by standard SU(2) approaches at the linear level. Rather, the physics is more conveniently described in terms of generalized SU(N) spin-wave theory, where the low-energy modes are described by $N - 1$ distinct bosons[18]. For example, anisotropic $S = 1$ systems where both transverse and longitudinal modes are expected, have been previously treated by linear SU(3) theories[17–23]. While linear SU(N) approaches to capture the correct number of low-energy modes, they are unable to reproduce the quasi-particle decay and renormalization generated by the interaction between these modes. To capture these effects requires going beyond the linear level and thus an objective of this paper is to generalize the $1/S$-expansion of the SU(2) treatment to SU(3) in order to account for the quasi-particle decay and renormalization produced by the interaction (nonlinear) terms using the quintessential example of interacting longitudinal and transverse modes for an $S = 1$ easy-plane quantum magnet as a test case.

In easy-plane quantum magnets, phase transitions can be driven by either fluctuations of the phase or the amplitude of the order parameter[24]. The phase fluctuations are the transverse modes of the order parameter (Goldstone modes in the long-wavelength limit), whereas amplitude fluctuations correspond to the longitudinal modes. Due to the gapless nature of the Goldstone transverse modes, the longitudinal or "Higgs" mode is kinematically allowed to decay into two transverse modes. This decay becomes more significant in low-dimensional systems. Indeed, the longitudinal mode in two-dimensional (2D) antiferromagnets was originally assumed to be overdamped due to an infrared divergence of the imaginary part of the longitudinal susceptibility[25,26]. However, more recent theoretical work predicted that the longitudinal peak should remain visible even in 2D[27–33]. One aspect of this problem, which has not been emphasized in previous works, is that the rather strong decay of the longitudinal mode is accompanied by a significant renormalization of the gap and the dispersion of the modes. As noted above, this additional many-body effect provides a hard test for theories that attempt to reproduce the measured decay of the Higgs mode.

As a starting point to understand the physics described above, we focus on the quasi-2D Heisenberg square lattice with effective $S = 1$ with an antiferromagnetic exchange coupling ($\widetilde{J}$) and a strong easy-plane single-ion anisotropy ($\widetilde{D}$). In this case, $\alpha = \widetilde{J}/\widetilde{D}$ can be viewed as a tuning parameter that can be used to drive a system from a quantum paramagnet (QPM) to an antiferromagnet (AFM) with an intervening QCP as shown in Fig. 1. Near the QCP, spontaneous symmetry breaking produces two transverse modes (one of them is a Goldstone mode) and a longitudinal Higgs mode. The longitudinal mode is unstable with respect to decay into a pair of transverse modes resulting in an intrinsic line broadening[9,34].

In this paper, we use inelastic neutron scattering to study the spin excitation spectrum of $Ba_2FeSiO_7$. The high-quality neutron-scattering data reveals a complex spectrum where transverse modes are resolution limited, whereas a longitudinal mode displays significant **Q**-dependent broadening throughout the Brillouin zone (BZ), demonstrating the importance of quasi-particle decay even away from the long-wavelength limit. The neutron-scattering results further show that the longitudinal mode has a very small gap clearly demonstrating that $Ba_2FeSiO_7$ is relatively close to a QCP. To understand the inelastic neutron-scattering data, we implement a generalized SU(3) spin-wave calculation[17,18,22] and compute the low-energy excitation spectrum of an effective low-energy spin $S = 1$ model. After demonstrating that the generalization of the well-known $1/S$-expansion of the SU(2) spin-wave theory[35–41] is simply a loop expansion[42] of the SU(3) spin-wave theory, we show that the one-loop correction is enough to account for the broadening of the longitudinal mode and the large renormalization of the gap and the dispersion of this mode. We further show that not including the one-loop corrections results in Hamiltonian parameters that place the exact ground state of the spin Hamiltonian for $Ba_2FeSi_2O_7$ on the nonmagnetic side of the QCP—contrary to experimental fact. This provides a dramatic demonstration of the importance of including renormalization effects, where the linear spin-wave calculation overestimates the stability range of the

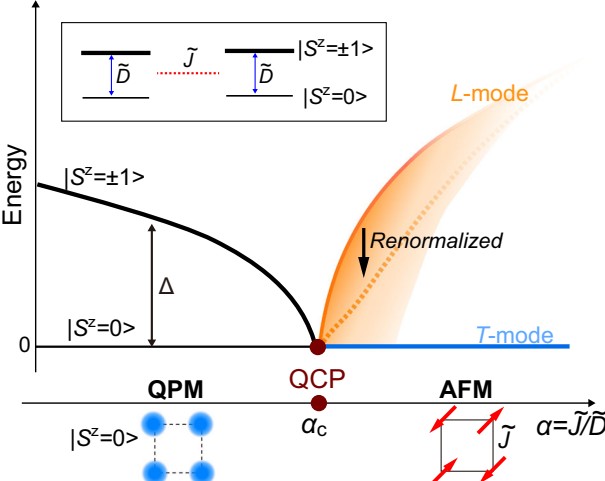

**Fig. 1 Schematic diagrams near the quantum critical point.** The schematic phase diagram illustrates the O(2) quantum critical point (QCP) between the antiferromagnetic (AFM) state and the quantum paramagnet (QPM) as a function of $\alpha = \widetilde{J}/\widetilde{D}$ ($\widetilde{J}$ is a Heisenberg exchange and $\widetilde{D}$ is a easy-plane single-ion anisotropy of effective $S = 1$). The low-energy excitations of the QPM are two degenerate $S^z = \pm 1$ modes (black line) with a gap, $\Delta$, which closes at the QCP. The spontaneous U(1) symmetry breaking leads to a gapless magnon or transverse mode ($T$-mode), indicated with a blue line, which is accompanied by a gapped longitudinal mode ($L$-mode) indicated with the orange line. Near the QCP, the energy and the lifetime of the $L$-mode are strongly renormalized (dashed orange line) due to the decay into the continuum of two transverse modes (shaded orange region).

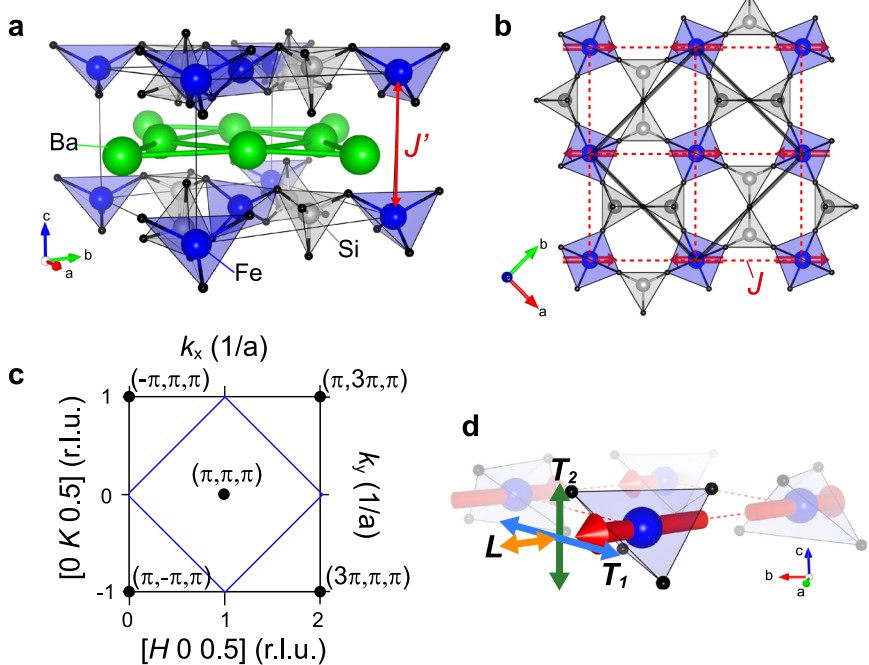

**Fig. 2 Crystal and magnetic structure of Ba$_2$FeSi$_2$O$_7$. a** Crystal structure of Ba$_2$FeSi$_2$O$_7$. Ba atoms separate layers composed of FeSi$_2$O$_7$, rendering a quasi-two-dimensional structure. **b** In the FeSi$_2$O$_7$ layer, FeO$_4$ tetrahedra are connected via SiO$_4$ polyhedra, and the adjacent two Fe$^{2+}$ atoms are exchange coupled by two oxygen ligands. The red dashed line indicates the exchange pathway $J$ within two-dimensional square spin network. The interlayer coupling $J'$ is found here to be much weaker than $J$. Red arrows indicate the moment direction in the collinear AFM phase as determined in ref. [44]. The black solid line indicates the chemical unit cell. **c** $HK$-reciprocal space with $L = 0.5$ in the tetragonal structure ($P\bar{4}2_1m$). The blue solid line and the black circle indicate the Brillouin zone and zone center, respectively. The coordinates ($H$, $K$, $L$) of the reciprocal lattice of the origin lattice are related to ($k_x$, $k_y$, $k_z$) of the magnetic lattice formed by the Fe$^{2+}$ atoms through $k_x = \pi(H - K)$, $k_y = \pi(H + K)$, and $k_z = 2\pi L$. **d** Illustration of the spin fluctuation modes. $T_1$ and $T_2$ indicate transverse fluctuation in the $ab$-plane and out-of the plane, respectively. $L$ indicates longitudinal fluctuation of spin.

magnetically ordered state. The fact that the one-loop correction can simultaneously account for the real and imaginary part of the self-energy of the longitudinal mode, as well as of the renormalization the transverse mode dispersion, confirms that the easy-plane quantum magnet Ba$_2$FeSi$_2$O$_7$ is an ideal platform for studying many-body effects in the proximity of the O(2) QCP.

## Results

**Model material**. Figure 2a illustrates the crystal structure of Ba$_2$FeSi$_2$O$_7$ comprising layers of FeSi$_2$O$_7$ separated by Ba atoms. As shown in Fig. 2b, the FeO$_4$ tetrahedra of the FeSi$_2$O$_7$ layer are connected via SiO$_4$ polyhedra and the two adjacent Fe$^{2+}$ atoms are coupled through the superexchange interaction, $J$, that is mediated by the two oxygen ligands (red dashed line in Fig. 1b). The resulting square lattice of magnetic moments is vertically stacked along the $c$ axis, leading to a quasi-2D simple tetragonal spin lattice.

A detailed description of the single-ion state of the Fe$^{2+}$ ion is given in Note 1 of the Supplementary Information. The combination of a relatively large spin–orbit coupling ($\lambda \sim 20$ meV) and a dominant tetrahedral crystal field ($\Delta_{Td}$), splits the free-ion levels, $^5D$ ($L = 2$, $S = 2$), into several multiplets. The lowest energy $S = 2$ multiplet has a significant orbital character due to the finite spin–orbit coupling, that combined with the tetragonal distortion ($\delta_{Tetra}$) by large compression of the FeO$_4$ tetrahedra leads to a rather strong easy-plane single-ion anisotropy[43,44]. The five $S = 2$ energy levels are then split into a singlet $S^z = 0$ ground state and two excited $S^z = \pm 1$ and $S^z = \pm 2$ doublets with energies $D$ and $4D$, respectively (see Fig. S1a of the Supplementary Information). Because the gap $D$ of the $S^z = \pm 1$ doublet is four times smaller than the gap of the $S^z = \pm 2$ doublet and the dominant super-exchange interaction $J$ is smaller than $D/4$ in Ba$_2$FeSi$_2$O$_7$, the low-

energy spectrum is well captured by projecting the $S = 2$ spin Hamiltonian into the $S^z = 0$ and $S^z = \pm 1$ low-energy states.

The resultant $S = 1$ effective spin Hamiltonian describes the competition between a QPM ($\tilde{J} \ll \tilde{D}$) with each spin of the lattice having dominant $S^z = 0$ character, and a collinear AFM state ($\alpha = \tilde{J}/\tilde{D} > \alpha_c$) with staggered magnetization in the $ab$ plane (see Fig. 2b). Ba$_2$FeSi$_2$O$_7$ turns out to be on the antiferromagnetic side with a Néel temperature $T_N = 5.2$ K[44]. Below $T_N$, the spins order antiferromagnetically with propagation vector $\mathbf{Q}_m = (1, 0, 0.5)$, corresponding to $(\pi, \pi, \pi)$ as shown in Fig. 2c. The magnetic moments are highly confined in the $ab$ plane due to easy-plane anisotropy, giving rise to the magnetic structure shown in Fig. 2b. A neutron diffraction study on a powder sample revealed a significantly reduced ordered moment of 2.95 $\mu_B$, which is only 63% of the full moment of 4.36 $\mu_B$ ($g_{ab} = 2.18$) expected for an $S = 2$ spin[44], suggesting the proximity to the quantum critical point. In addition, as described in further detail below, our analysis confirms that $\alpha = \tilde{J}/\tilde{D} \sim 0.184$ is close to the critical value, $\alpha_c^{2D} = 0.18$ and $\alpha_c^{3D} = 0.1$ for 2D and 3D, respectively, obtained from quantum Monte Carlo simulations[22].

The spin excitations of Ba$_2$FeSi$_2$O$_7$ are generically described by an antiferromagnetic $S = 2$ spin Hamiltonian on a simple tetragonal lattice:

$$
\begin{aligned}
\mathcal{H} = &J \sum_{\langle r,r' \rangle} [S_r^x S_{r'}^x + S_r^y S_{r'}^y + \Delta S_r^z S_{r'}^z] \\
&+ J' \sum_{\langle\langle r,r' \rangle\rangle} [S_r^x S_{r'}^x + S_r^y S_{r'}^y + \Delta' S_r^z S_{r'}^z] \\
&+ D \sum_r (S_r^z)^2.
\end{aligned} \tag{1}
$$

The bracket $\langle \mathbf{r},\mathbf{r}' \rangle (\langle\langle \mathbf{r},\mathbf{r}' \rangle\rangle)$ indicates that the sum runs over intralayer (interlayer) nearest-neighbor spins with isotropic

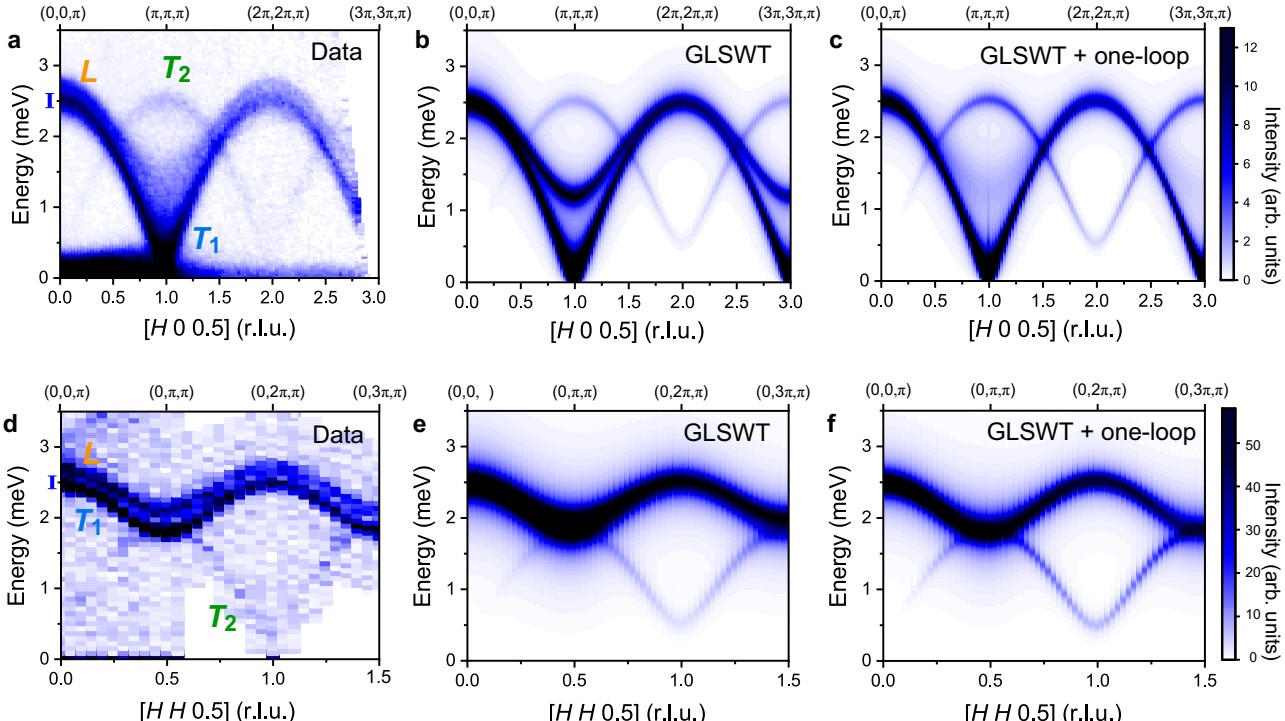

**Fig. 3 Inelastic neutron scattering of Ba$_2$FeSi$_2$O$_7$. a** Contour map of the inelastic neutron-scattering (INS) data as a function of energy and momentum transfer along the [H, 0, 0.5] direction measured at $T = 1.6$ K ($<T_N$) using the HYSPEC time-of-flight spectrometer at SNS. **d** Contour map of the INS data as a function of energy and momentum transfer along [H, H, 0.5] direction measured at $T = 1.4$ K ($<T_N$) using the cold Neutron Triple-Axis spectrometer (CTAX) at HFIR. The instrumental resolutions at energy = 2.5 meV for each instrument are indicated with blue bars along the y-axis in **a** and **d**. The two transverse modes and the longitudinal mode are labeled with $T_1$, $T_2$, and $L$, respectively. **b**, **c**, **e**, **f** INS intensities calculated by the generalized linear spin-wave theory (GLSWT) and GLSWT plus one-loop corrections (GLSWT+one-loop) with the parameter sets $\mathcal{A}$ and $\mathcal{B}$ given in Table 1, respectively. The instrumental resolution of HYSPEC and CTAX was modeled in the calculated spectra using a Lorentzian function.

superexchange interaction $J(J')$. $\Delta(\Delta')$ is the intralayer (interlayer) uniaxial anisotropy and the last term represents the easy-plane single-ion anisotropy ($D > 0$).

In the large $D/J$ limit, the $S^z = \pm 2$ doublet is separated from the $S^z = \pm 1$ doublet by an energy gap $3D$. The low-energy subspace of magnetic excitations can then be further reduced by projecting out the $S^z = \pm 2$ doublet. The reduced low-energy Hamiltonian $\mathcal{H}_{eff}$ results from projecting $\mathcal{H}$ onto the low-energy subspace $S$ spanned by the triplet of states with $S^z = 0, \pm 1$: $\mathcal{H}_{eff} = \mathcal{P}_S \mathcal{H} \mathcal{P}_S$. The resulting effective spin $S = 1$ Hamiltonian is

$$
\begin{aligned}
\mathcal{H}_{eff} = &\tilde{J} \sum_{\langle \mathbf{r},\mathbf{r}'\rangle} [s_\mathbf{r}^x s_{\mathbf{r}'}^x + s_\mathbf{r}^y s_{\mathbf{r}'}^y + \tilde{\Delta} s_\mathbf{r}^z s_{\mathbf{r}'}^z] \\
&+ \tilde{J}' \sum_{\langle\langle \mathbf{r},\mathbf{r}'\rangle\rangle} [s_\mathbf{r}^x s_{\mathbf{r}'}^x + s_\mathbf{r}^y s_{\mathbf{r}'}^y + \tilde{\Delta}' s_\mathbf{r}^z s_{\mathbf{r}'}^z] \\
&+ \tilde{D} \sum_\mathbf{r} (s_\mathbf{r}^z)^2 .
\end{aligned}
\tag{2}
$$

with $\tilde{J} = 3J$, $\tilde{J}' = 3J'$, $\tilde{\Delta} = \Delta/3$, $\tilde{\Delta}' = \Delta'/3$, and $\tilde{D} = D$. As we will see below, this simple effective Hamiltonian can explain not only the in-plane antiferromagnetic ordering observed in Ba$_2$FeSi$_2$O$_7$ (see Fig. 2b), but also the spectra of quasi-particle excitations, including rather strong renormalization effects due to proximity to the QCP.

**Inelastic neutron scattering.** To investigate the spin excitation spectrum in Ba$_2$FeSi$_2$O$_7$, we performed inelastic neutron scattering using two instruments; the cold neutron triple-axis spectrometer (CTAX) at the High Flux Isotope Reactor, and the time-of-flight hybrid spectrometer (HYSPEC) at the Spallation Neutron Source at Oak Ridge National Laboratory[45]. An overview of the inelastic neutron-scattering results is presented in Fig. 3 through contour

maps of the neutron-scattering intensity, $I(\mathbf{Q}, \omega)$, along [H, H, 0.5] and [H, 0, 0.5]. For both spectra, strongly dispersive spin excitations extending up to energy ~2.7 meV are observed. Whereas the dispersion along [0, 0, L] direction is weak with a bandwidth of ~0.5 meV (see Note 4 in the Supplementary Information), which is expected for spin excitations of a quasi-two-dimensional spin system.

There are several distinct features in the inelastic neutron-scattering data. An intense spin-wave excitation emanates from the magnetic zone center (ZC), $\mathbf{Q} = (1, 0, 0.5)$, which arises due to the in-phase oscillation between Fe$^{2+}$ spins in the plane. We refer to this mode as $T_1$. Along the [H, 0, 0.5] direction toward the zone boundary (ZB) at $\mathbf{Q} = (0, 0, 0.5)$, the $T_1$-mode reaches its maximum energy of ~2.5 meV. Another weak, but sharp mode, is visible along [H, 0, 0.5] with an energy of 2.5 meV at the ZC. We refer to this mode as $T_2$. These two modes are expected for a strong easy-plane antiferromagnet, where transverse magnons split into gapless in-plane fluctuations ($T_1$-mode) and gapped out-of-plane fluctuations ($T_2$-mode). The finite value of the energy gap of the out-of-plane fluctuation at the ZC is associated with the strength of the easy-plane single-ion anisotropy[46].

The $T_1$ and $T_2$ transverse magnon modes are also observed along the [H, H, 0.5] direction in Fig. 3d. Noticeably, an additional sharp mode is observed at the top of the $T_1$-mode. This mode is visible along the entire Brillouin zone boundary. We refer to this additional mode as "$L$"-mode. The $L$-mode is visible in the spectra along [H, 0, 0.5] as well, however, it exhibits dramatic line broadening near the ZC. To demonstrate more clearly the $\mathbf{Q}$-dependence of the modes, Fig. 4 shows cuts at constant momentum transfers for multiple points along [H, 0, 0.5] and [H, H, 0.5]. Two pronounced peaks, corresponding to the $T_1$- and $L$-modes, remain sharp along

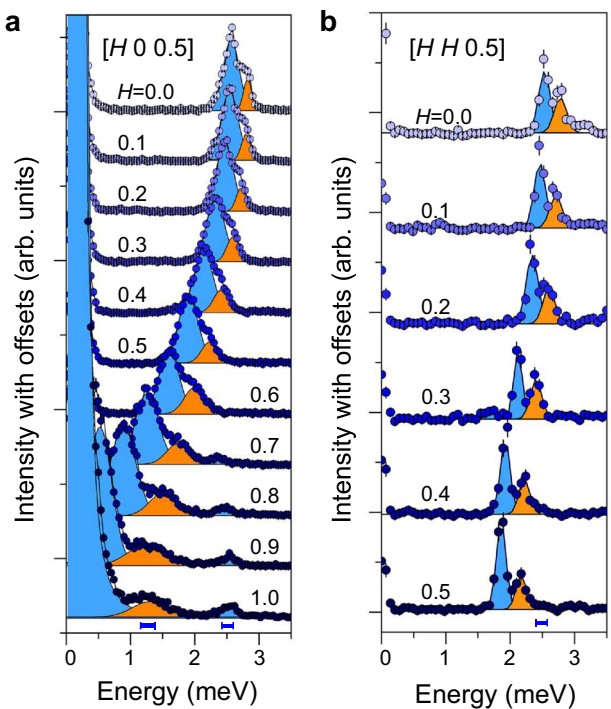

**Fig. 4 Detailed line-cuts of INS spectra. a** Constant momentum cuts at points along the [H, 0, 0.5] direction measured using HYSPEC at SNS, integrated over $H = [H-0.05, H+0.05]$ at selected $H$, $K = [-0.1, 0.1]$, and $L = [0.4, 0.6]$. **b** Constant momentum cuts at points along the [H, H, 0.5] direction measured using CTAX at HFIR. Blue bars at the bottom of the panels indicate the instrumental resolutions for HYSPEC and CTAX at the proximate energy transfers. The blue and orange shaded regions are the results of fitting Gaussian line shape to transverse ($T_1$, $T_2$) and longitudinal (L) modes, respectively.

the ZB (Fig. 4b). As already noted, the situation is very different near the ZC where the L-mode is significantly broadened (Fig. 4a). We note that the L-mode remains a broad peak near the ZC, rather than a featureless excitation. To investigate the extent of the broadening effect, Gaussian line shapes for the $T_1$-, $T_2$-, and L-modes were fit to the individual cuts in Fig. 4. The line widths obtained from the fits are displayed in Fig. 7a–d. These results reveal that the L-mode is three times broader than the instrumental resolution at the ZC (see Fig. 4a), whereas it has comparable line width to instrumental resolution near the ZB.

**Generalized spin waves**. In this section, we introduce a generalized SU(3) spin-wave calculation[17,18,22,47], which is required to capture the two low-energy (longitudinal and transverse) modes of $Ba_2FeSi_2O_7$. Clearly, a linear treatment is not enough to capture the decay of the longitudinal mode into two transverse modes. Consequently, the main aim of this section is to lay the groundwork for introducing the loop expansion[42] (generalization of the 1/S-expansion[35–41]) in the section describing the nonlinear corrections.

To account for the transverse and longitudinal modes revealed by the INS experiment, the usual SU(2) spin-wave theory (SWT) must be generalized to SU(3)[18], by introducing the SU(3) Schwinger boson representation of the spin operators $S_{\mathbf{r}}^\nu = \boldsymbol{b}_{\mathbf{r}}^\dagger \mathcal{S}^\nu \boldsymbol{b}_{\mathbf{r}}$, where $\boldsymbol{b}_{\mathbf{r}} = (b_{\mathbf{r},+1}, b_{\mathbf{r},-1}, b_{\mathbf{r},0})^T$,

$$\mathcal{S}^x = \frac{1}{\sqrt{2}}(\lambda_4 + \lambda_6), \quad \mathcal{S}^y = \frac{1}{\sqrt{2}}(\lambda_5 - \lambda_7), \quad \mathcal{S}^z = \lambda_3, \quad (3)$$

$\lambda_i$ are the Gel–Mann matrices and the Schwinger boson operators satisfy the local constraint

$$\sum_{m=\pm 1,0} b_{\mathbf{r},m}^\dagger b_{\mathbf{r},m} = M = 1. \quad (4)$$

We note that the SU(3) Schwinger boson representation of the spin operators should not be confused with the Schwinger boson approximation[36,48–50], which is qualitatively different from the semi-classical approach that we describe below. The magnetically ordered state of $Ba_2FeSi_2O_7$ can be approximated by a product (mean-field) state of normalized SU(3) coherent states

$$|\psi_{\mathbf{r}}\rangle = \cos\theta|0\rangle + (\sin\theta\cos\phi|1\rangle + \sin\theta\sin\phi|-1\rangle)e^{i\mathbf{Q_m} \cdot \mathbf{r}}, \quad (5)$$

where $\mathbf{Q_m} = (\pi, \pi, \pi)$ ((1, 0, 0.5) in the chemical lattice) is the AFM ordering wave vector. Although a general SU(3) coherent state is parameterized by four independent parameters for degenerate representations[51], the two independent parameters $\theta$ and $\phi$ are enough to describe the collinear order under consideration. The three basis states $|m\rangle(m = 0, \pm1)$ are represented by creating a boson with the quantum number $m$ from the vacuum: $|m\rangle = b_{\mathbf{r},m}^\dagger|\phi\rangle$.

As in the usual spin-wave theory, we introduce an SU(3) transformation that rotates the boson operators, $\widetilde{\boldsymbol{b}}_{\mathbf{r}} = U_{\mathbf{r}}\boldsymbol{b}_{\mathbf{r}}$, to a local basis that includes the coherent SU(3) state (5) as one of its three elements. This local transformation allows us to align the quantization axis with the direction of the local SU(3) order parameter. The spatial dependence of $U_{\mathbf{r}}$ can be removed by working in a twisted frame, where the original AFM order becomes a FM one. This can be done by rotating the spin reference frame of one of the two sublattices of the tetragonal lattice by an angle $\pi$ along the $z$ axis: $s_{\mathbf{r}}^z \to s_{\mathbf{r}}^z$, and $s_{\mathbf{r}}^{x,y} \to -s_{\mathbf{r}}^{x,y}$. In the new reference frame, the effective Hamiltonian (2) becomes

$$\tilde{\mathcal{H}}_{\text{eff}} = \tilde{J}\sum_{\langle\mathbf{r},\mathbf{r}'\rangle,\nu} a_\nu s_{\mathbf{r}}^\nu s_{\mathbf{r}'}^\nu + \tilde{J}'\sum_{\langle\mathbf{r},\mathbf{r}'\rangle,\nu} b_\nu s_{\mathbf{r}}^\nu s_{\mathbf{r}'}^\nu + \tilde{D}\sum_{\mathbf{r}}(s_{\mathbf{r}}^z)^2, \quad (6)$$

with $a_x = a_y = b_x = b_y = -1$, $a_z = \widetilde{\Delta}$ and $b_z = \widetilde{\Delta}'$, and the SU(3) transformation reads

$$U = \begin{pmatrix} -\sin\phi & \cos\phi & 0 \\ \cos\theta\cos\phi & \cos\theta\sin\phi & -\sin\theta \\ \sin\theta\cos\phi & \sin\theta\sin\phi & \cos\theta \end{pmatrix}. \quad (7)$$

The bosonic representation of $\widetilde{\mathcal{H}}_{\text{eff}}$ is

$$\tilde{\mathcal{H}}_{\text{eff}} = \tilde{J}\sum_{\langle\mathbf{r},\mathbf{r}'\rangle,\nu} a_\nu \tilde{\boldsymbol{b}}_{\mathbf{r}}^\dagger \widetilde{\mathcal{S}}^\nu \tilde{\boldsymbol{b}}_{\mathbf{r}} \tilde{\boldsymbol{b}}_{\mathbf{r}'}^\dagger \widetilde{\mathcal{S}}^\nu \tilde{\boldsymbol{b}}_{\mathbf{r}'}$$
$$+ \tilde{J}'\sum_{\langle\mathbf{r},\mathbf{r}'\rangle,\nu} b_\nu \widetilde{\boldsymbol{b}}_{\mathbf{r}}^\dagger \widetilde{\mathcal{S}}^\nu \widetilde{\boldsymbol{b}}_{\mathbf{r}} \widetilde{\boldsymbol{b}}_{\mathbf{r}'}^\dagger \widetilde{\mathcal{S}}^\nu \widetilde{\boldsymbol{b}}_{\mathbf{r}'} \quad (8)$$
$$+ \tilde{D}\sum_{\mathbf{r}}(1 - \tilde{\boldsymbol{b}}_{\mathbf{r}}^\dagger \widetilde{\mathcal{A}}\tilde{\boldsymbol{b}}_{\mathbf{r}}),$$

where $\widetilde{\mathcal{S}}^\nu = U\mathcal{S}^\nu U^\dagger$, $\widetilde{\mathcal{A}} = U\mathcal{A}U^\dagger$, and $\mathcal{A}_{\alpha\beta} = \delta_{\alpha,0}\delta_{\beta,0}$. Note that the unitary transformation (7) is chosen in such a way that the $\widetilde{b}_{\mathbf{r},0}$ boson is macroscopically occupied, namely $\langle\widetilde{b}_{\mathbf{r},0}\rangle = \langle\widetilde{b}_{\mathbf{r},0}^\dagger\rangle \simeq \sqrt{M}$. According to the constraint (4), $M = 1$ for the case of interest. However, we will keep using $M$ because $1/M$ is the parameter of the perturbative expansion that will be introduced below. Note that $M = 2S$ for the usual SU(2) spin-wave theory. The main assumption behind the $1/M$ expansion is that $\langle\widetilde{b}_{\mathbf{r},-1}^\dagger\widetilde{b}_{\mathbf{r},-1}\rangle, \langle\widetilde{b}_{\mathbf{r},+1}^\dagger\widetilde{b}_{\mathbf{r},+1}\rangle \ll M$. Under this assumption, we can expand the spin operators $S^\mu$ and the quadrupolar operator $(S^z)^2$ in powers of $1/M$ (see Note 5 in

the Supplementary Information). The resulting expansion of $\widetilde{\mathcal{H}}_{\text{eff}}$ is

$$\widetilde{\mathcal{H}}_{\text{eff}} = M^2 \mathcal{H}^{(0)} + M \mathcal{H}^{(2)} + M^{1/2} H^{(3)} + M^0 H^{(4)} + O(M^{-1}), \quad (9)$$

where the linear term $H^{(1)}$ vanishes because the parameters $\theta$ and $\phi$ in Eq. (5) are determined by minimizing the mean-field energy

$$\mathcal{H}^{(0)} = (2\tilde{J}\tilde{\Delta} + \tilde{J}'\tilde{\Delta}')\sin^4\theta\cos^2 2\phi$$

$$- \frac{1}{2}(2\tilde{J} + \tilde{J}')\sin^2 2\theta(1 + \sin 2\phi) + \tilde{D}\sin^2\theta. \quad (10)$$

Since the AFM order is invariant under time reversal followed by one lattice translation, the states $S^z = \pm 1$ must have equal weight in the mean-field state (5), implying that $\phi = \pi/4$. By minimizing $\mathcal{H}^{(0)}$ with respect to $\theta$, we obtain

$$x \equiv \sin^2\theta = \frac{1}{2} - \frac{\tilde{D}}{8(2\tilde{J} + \tilde{J}')}. \quad (11)$$

The quadratic term $\mathcal{H}^{(2)}$ represents the generalized linear spin-wave (GLSW) Hamiltonian. After Fourier transforming the bosonic operators,

$$\tilde{b}_{\mathbf{r}\alpha} = \frac{1}{\sqrt{N_s}} \sum_{\mathbf{k}} \tilde{b}_{\mathbf{k}\alpha} e^{i\mathbf{k}\cdot\mathbf{r}}, \quad (12)$$

where $N_s$ is the number of sites, $\mathcal{H}^{(2)}$ can be brought into a compact form by introducing the Nambu spinor $\vec{b}_{\mathbf{k}} = (\tilde{b}_{\mathbf{k},+1}, \tilde{b}_{\mathbf{k},-1}, \tilde{b}^{\dagger}_{-\mathbf{k},+1}, \tilde{b}^{\dagger}_{-\mathbf{k},-1})^T$,

$$\mathcal{H}^{(2)} = \sum_{\mathbf{k}} \sum_{\alpha,\beta=\pm 1} \vec{b}^{\dagger}_{\mathbf{k}} \mathcal{H}^{(2)}(\mathbf{k}) \vec{b}_{\mathbf{k}}, \quad (13)$$

with

$$\mathcal{H}^{(2)}(\mathbf{k}) = \begin{pmatrix} \Delta_{\alpha\beta}(\mathbf{k}) & \Lambda_{\alpha\beta}(\mathbf{k}) \\ \Lambda_{\beta\alpha}(\mathbf{k}) & \Delta_{\beta\alpha}(\mathbf{k}) \end{pmatrix}. \quad (14)$$

The matrix elements are

$$\Delta_{\alpha\beta}(\mathbf{k}) = \sum_{\nu} \Big[ (2a_{\nu}\tilde{J} + b_{\nu}\tilde{J}')(\tilde{\mathcal{S}}^{\nu}_{\alpha\beta}\tilde{\mathcal{S}}^{\nu}_{00} - (\tilde{\mathcal{S}}^{\nu}_{00})^2\delta_{\alpha\beta})$$

$$+ (\tilde{J}a_{\nu}\sum_{\nu'=x,y}\cos k_{\nu'} + \tilde{J}'b_{\nu}\cos k_z)\tilde{\mathcal{S}}^{\nu}_{\alpha 0}\tilde{\mathcal{S}}^{\nu}_{0\beta}\Big] \quad (15)$$

$$- \frac{\tilde{D}}{2}(\tilde{\mathcal{A}}_{\alpha\beta} - \tilde{\mathcal{A}}_{00}\delta_{\alpha\beta}),$$

$$\Lambda_{\alpha\beta}(\mathbf{k}) = \sum_{\nu} \tilde{\mathcal{S}}^{\nu}_{\alpha 0}\tilde{\mathcal{S}}^{\nu}_{\beta 0}\Big[ \tilde{J}a_{\nu}\sum_{\nu'=x,y}\cos k_{\nu'} + \tilde{J}'b_{\nu}\cos k_z\Big]. \quad (16)$$

The collinear mean-field state (5) has a residual $Z_2$ symmetry associated with a $\pi$-rotation along the direction of the ordered moments (local $\tilde{z}$ axis). The bosonic operator $\tilde{b}^{\dagger}_{+1}$ picks up minus sign under this $Z_2$ symmetry because it creates the state with $\tilde{S}^z = -1$. In contrast, the bosonic operator $\tilde{b}^{\dagger}_{-1}$ remains invariant because it creates the state with $\tilde{S}^z = 0$. This symmetry analysis implies that the $\tilde{b}_{+1}$ and $\tilde{b}_{-1}$ bosons must be decoupled in $\mathcal{H}^{(2)}$ because a non-vanishing hybridization term would otherwise break this $Z_2$ symmetry:

$$\mathcal{H}^{(2)} = \sum_{\mathbf{k},\alpha=\pm 1} [A_{\mathbf{k},\alpha}\tilde{b}^{\dagger}_{\mathbf{k},\alpha}\tilde{b}_{\mathbf{k},\alpha} - \frac{B_{\mathbf{k},\alpha}}{2}(\tilde{b}_{-\mathbf{k},\alpha}\tilde{b}_{\mathbf{k},\alpha} + \tilde{b}^{\dagger}_{\mathbf{k},\alpha}\tilde{b}^{\dagger}_{-\mathbf{k},\alpha})] \quad (17)$$

with $\gamma^{xy}_{\mathbf{k}} = \cos(k_x) + \cos(k_y)$, $\gamma^z_{\mathbf{k}} = \cos(k_z)$ and the expressions for $A_{\mathbf{k},\alpha}$ and $B_{\mathbf{k},\alpha}$ are given in Note 5 of the Supplementary Information.

The diagonal form of $\mathcal{H}^{(2)}$,

$$\mathcal{H}^{(2)} = \sum_{\mathbf{k},\alpha=\pm 1} \omega_{\mathbf{k},\alpha}\left(\beta^{\dagger}_{\mathbf{k},\alpha}\beta_{\mathbf{k},\alpha} + \frac{1}{2}\right) \quad (18)$$

is then obtained by applying an independent Bogoliubov transformation for each bosonic flavor,

$$\tilde{b}_{\mathbf{k},\pm 1} = u_{\mathbf{k},\pm 1}\beta_{\mathbf{k},\pm 1} + v_{\mathbf{k},\pm 1}\beta^{\dagger}_{-\mathbf{k},\pm 1}, \quad (19)$$

with

$$u_{\mathbf{k},\pm 1} = \sqrt{\frac{1}{2}\left(\frac{|A_{\mathbf{k},\pm 1}|}{\omega_{\mathbf{k},\pm 1}} + 1\right)},$$

$$v_{\mathbf{k},\pm 1} = \frac{B_{\mathbf{k},\pm}}{|B_{\mathbf{k},\pm}|}\sqrt{\frac{1}{2}\left(\frac{|A_{\mathbf{k},\pm 1}|}{\omega_{\mathbf{k},\pm 1}} - 1\right)}. \quad (20)$$

The operators $\beta^{\dagger}_{\mathbf{k},\pm 1}$ create quasi-particles with energy

$$\omega_{\mathbf{k},\pm 1} = \sqrt{A^2_{\mathbf{k},\pm 1} - B^2_{\mathbf{k},\pm 1}}, \quad (21)$$

where $\omega_{\mathbf{k},+1}(\omega_{\mathbf{k},-1})$ is the dispersion relation of the transverse (longitudinal) modes. The neutron-scattering intensity $I(\mathbf{Q},\omega)$ is related to the spin-spin correlation function through

$$I(\mathbf{Q},\omega) \propto f^2(\mathbf{Q})\sum_{\mu,\nu}\left(\delta_{\mu\nu} - \frac{\hat{Q}_{\mu}\hat{Q}_{\nu}}{Q^2}\right)$$

$$\times \frac{1}{2\pi N_s}\sum_{i,j}^{N_s}\int_{-\infty}^{+\infty}dt e^{i\omega t - i\mathbf{Q}\cdot(\mathbf{r}_i - \mathbf{r}_j)}\langle s^{\mu}_i(t)s^{\nu}_j(0)\rangle, \quad (22)$$

where $\mathbf{Q}$ is the momentum vector transfer, and $f(\mathbf{Q})$ is the magnetic form factor of $Fe^{2+}$. In the Discussion section, we will show that although the GLSW approach discussed in this section can reproduce the dispersion relations of all observed low-energy modes in $Ba_2FeSi_2O_7$, it cannot account for various interaction effects that are revealed by the INS experiments. To capture these effects, we must then include the next order terms in the $1/M$-expansion.

**Nonlinear corrections**. In this section, we demonstrate that the generalization of the $1/S$-expansion is simply a loop expansion. Based on this result, we compute the one-loop corrections to the linear theory presented in the previous section. As we explain in the next section, the one-loop correction accounts for both the broadening and the energy renormalization of the longitudinal mode near the zone center.

After Fourier transforming and applying a Bogoliubov transformation, the cubic contributions to the generalized spin-wave theory become

$$\mathcal{H}^{(3)} = \mathcal{H}^{(3)}_c + \mathcal{H}^{(3)}_l, \quad (23)$$

with

$$\mathcal{H}^{(3)}_c = \frac{1}{\sqrt{N_s}}\sum_{\mathbf{q}_i}\sum_{\alpha_i=\pm 1}\delta(\mathbf{q}_1 + \mathbf{q}_2 + \mathbf{q}_3)$$

$$\times \Big[\frac{1}{3!}V^{(3)}_s(\mathbf{q}_{1,2,3},\alpha_{1,2,3})\beta_{\mathbf{q}_1,\alpha_1}\beta_{\mathbf{q}_2,\alpha_2}\beta_{\mathbf{q}_3,\alpha_3}$$

$$+ \frac{1}{2!}V^{(3)}_d(\mathbf{q}_{1,2,3},\alpha_{1,2,3})\beta^{\dagger}_{\mathbf{q}_1,\alpha_1}\beta^{\dagger}_{\mathbf{q}_2,\alpha_2}\beta_{\mathbf{q}_3,\alpha_3} + h.c.\Big], \quad (24)$$

and

$$
\begin{aligned}
\mathcal{H}_l^{(3)} &= \frac{1}{\sqrt{N_s}} \sum_{\mathbf{q}} \sum_{\alpha=\pm 1} [V_l^{(3)}(\mathbf{q}, 0, \mathbf{q}; \alpha, -1, \alpha)\beta_{0,-1}^\dagger + h.c.] \\
&= \sqrt{N_s} \sum_{\alpha=\pm 1} [V_{L,\alpha}\beta_{0,-1}^\dagger + h.c.].
\end{aligned}
\tag{25}
$$

Here $V_d^{(3)}$ and $V_s^{(3)}$ are the decay and sink vertices, respectively. The symmetry-allowed cubic vertices are depicted in the second and third lines of Fig. 4. Note that, unlike the SU(2) case, collinear magnetic ordering does not preclude cubic terms in the expansion (9) of the generalized SU($N$) spin-wave theory with $N > 2$. For the SU(3) case under consideration, the residual $Z_2$ symmetry ($\pi$-rotation along the local $\widetilde{z}$-axis) only requires that the $\widetilde{b}_{+1}$ boson must appear an even number of times (e.g., $\widetilde{b}_{+1}\widetilde{b}_{+1}$ or $\widetilde{b}_{+1}^\dagger\widetilde{b}_{+1}^\dagger$) in the cubic terms. $\mathcal{H}_l^{(3)}$ is a linear term that originates from the normal ordering of the cubic vertices. This term renormalizes the optimal value $\theta$ that was obtained from the minimization of $\mathcal{H}^{(0)}$. The integral of $V_l^{(3)}(\mathbf{q};\alpha, -1)$ over the entire Brillouin zone is the so-called cubic–linear vertex, which is nonzero only for the longitudinal boson at the ordering wave vector $\mathbf{q} = 0$ (in the twisted frame). The explicit forms of $V_{d,s}^{(3)}$ and $V_l^{(3)}$ are derived in Note 7 of the Supplementary Information.

We will now describe the construction of a systematic perturbative field theory that is controlled by $1/M$. This scheme can be applied to study anharmonicities starting from any generalized spin-wave theory based on a Schwinger boson representation of the generators of SU($N$). The well-known $1/S$-expansion will be recovered for the particular case $N = 2$ and $M = 2S$. As we will demonstrate below, the $1/M$-expansion is just a particular example of the loop expansion that is commonly used to describe spontaneous symmetry breaking in particle theory[42]. The connection is more evident after noticing that $M$ becomes an overall prefactor of the rescaled Hamiltonian (Eq. (9)), $H = H_{eff}/M$, once we also rescale the bosonic fields according to $\bar{b}_{\mathbf{r},\nu} = \widetilde{b}_{\mathbf{r},\nu}/\sqrt{M}$. Since the original interaction vertices $V^{(n)}(n \geq 3)$ scale as $V^{(n)} \sim (M)^{2-\frac{n}{2}}$, all vertices of the rescaled Hamiltonian $H(\bar{b}_{\mathbf{r},\nu}, b_{\mathbf{r},\nu}^\dagger)$ become of order $M$, while the propagator is still of order $1/M$. Thus, the order $p$ of a particular one-particle irreducible diagram is $V - I$, where $V$ is the number of vertices and $I$ is the number of internal lines (note that the frequency $\omega$ is of order $M^0$ because the quadratic contribution $\langle H^{(2)} \rangle$ is independent of $M$). Since the number of loops is $L = I - V + 1$ (Every vertex introduces a delta function that reduces the number of independent momenta by one, except for one delta function that is left over for overall energy-momentum conservation), we obtain the desired result: $p = 1 - L$.

Let us rederive this result without rescaling the fields and the Hamiltonian. As we already mentioned, Eq. (9) tells us that the interaction vertices $V^{(n)}(n \geq 3)$ scale as $V^{(n)} \sim (M)^{2-\frac{n}{2}}$. The quasi-particle propagator

$$
\mathcal{G}_{0,\alpha}(\mathbf{k}, i\omega) = (-i\omega + \omega_{\mathbf{k},\alpha})^{-1}, \quad \alpha = \pm 1
\tag{26}
$$

where $\omega$ is the Matsubara frequency, scales as $\mathcal{G}_{0,\alpha}(k) \sim M^{-1}$ because $\omega_{\mathbf{k},\alpha}$ is of order $M$ (see Eq. (9)). The dressed single-particle propagator is obtained from the Dyson equation,

$$
\mathcal{G}^{-1}(\mathbf{k}, i\omega) = \mathcal{G}_0^{-1}(\mathbf{k}, i\omega) - \Sigma(\mathbf{k}, i\omega),
\tag{27}
$$

where $\Sigma(\mathbf{k}, i\omega)$ is the single-particle self-energy. At a given order in $M$, the dressed propagator includes two external legs, $L$ independent loops, $I$ internal lines (bare propagators $\mathcal{G}_0$) and $V_n$ interaction vertices of the type $V^{(n)}$. After a summation over the

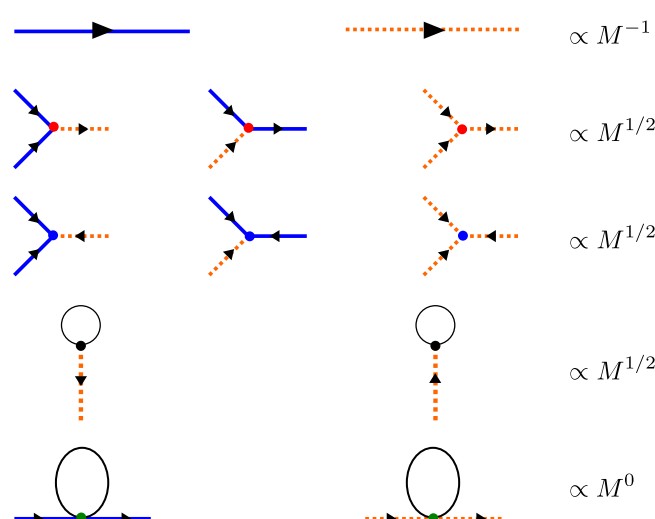

**Fig. 5 Basic ingredients of the perturbative field theory in $1/M$ for $Ba_2Fe_2Si_2O_7$.** Solid (dash) lines represent the bare propagator of the transverse (longitudinal) boson. The symmetry-allowed cubic vertices are shown on the second and third lines. The red (blue) dot represents a decay (sink) vertex. The cubic–linear vertices are listed on the fourth line. The last line represents the normal vertex $V_{\alpha\alpha}^{(4,N)}$ from $\mathcal{H}^{(4)}$.

Matsubara frequency $\omega \sim M^1$, each loop gives a contribution of order $M^1$. Hence, the order $p$ of a particular one-particle irreducible diagram contributing to $\Sigma(\mathbf{k}, i\omega)$ is

$$
p = L - I + \sum_{n \geq 3} V_n \left[ 2 - \frac{n}{2} \right].
\tag{28}
$$

Since each internal line connects a pair of vertices, we have

$$
\sum_{n \geq 3} nV_n = 2I + 2,
\tag{29}
$$

where $\sum_{n \geq 3} nV_n$ is the total number of lines. Furthermore, the number of loops is equal to the number of independent momentum integrals. From the conservation of momentum at each vertex, we have

$$
L = I - \left[ \sum_{n \geq 3} V_n - 1 \right].
\tag{30}
$$

By combining the above results, we obtain

$$
p = 1 + \sum_{n \geq 3} V_n - \sum_{n \geq 3} \frac{nV_n}{2} = 1 - L,
\tag{31}
$$

implying that the order of a given diagram is determined by the number of loops.

The lowest-order $\mathcal{O}(M^0)$ Feynman diagrams are shown in Fig. 6. Since the inverse of the bare boson propagator is of order $\mathcal{O}(M^1)$, the remaining diagrams of order $\mathcal{O}(M^0)$ give a relative $1/M$-correction to the poles of the bare propagators. The real part of the new poles corresponds to the renormalized single-particle energy, whereas the imaginary part corresponds to the decay rate, which is responsible for the broadening of the quasi-particle peaks measured with INS.

The contributions to the self-energy from the decay and from the source diagrams shown in Fig. 6 are

$$
\Sigma_\alpha^d(\mathbf{q}, i\omega) = \frac{1}{2N_s} \sum_{\mathbf{k},\alpha_1,\alpha_2=\pm 1} \frac{|V_d^{(3)}(\bar{\mathbf{k}}, \mathbf{k} + \bar{\mathbf{q}}, \mathbf{q}; \alpha_1, \alpha_2, \alpha)|^2}{i\omega - \omega_{\mathbf{k},\alpha_1} - \omega_{\mathbf{q}+\bar{\mathbf{k}},\alpha_2}},
\tag{32}
$$

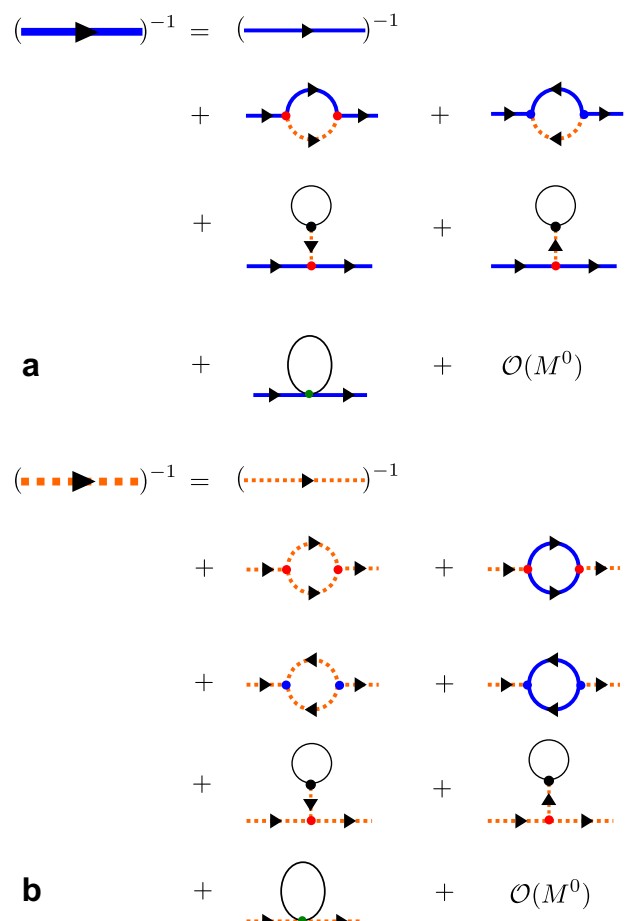

**Table 1 Parameter sets of GLSWT and GLSWT+one-loop models.**

| Theory | Label | $\widetilde{J}$ (meV) | $\widetilde{D}$ (meV) |
|---|---|---|---|
| GLSWT | $\mathcal{A}$ | 0.245 (7) | 1.61 (6) |
| GLSWT+one-loop | $\mathcal{B}$ | 0.266 (6) | 1.42 (4) |

The parameters of the effective $S = 1$ model extracted by fitting the Gaussian-peak centers of the experimental dispersion with the GLSWT and GLSWT + one-loop calculated energies at the zone center $\mathbf{Q_m} = (1, 0, 0.5)$. In both cases, we assume $\widetilde{J'} = 0.1\widetilde{J}$, and $\widetilde{\Delta} = \widetilde{\Delta}' = 1/3$, i.e. $\Delta = \Delta' = 1$ for the $S = 2$ model (Heisenberg model without exchange anisotropy). The parameter set is referred to by its label ($\mathcal{A}$ or $\mathcal{B}$) in the text.

**Fig. 6 Diagrammatic representation of the Dyson equation. a** One-loop diagrams that contribute up to the order $M^0$ for the transverse boson. **b** One-loop diagrams that contribute up to the order $M^0$ for the longitudinal boson. The dressed propagator is denoted by a thick line, whereas the bare propagator is denoted by a thin line.

and

$$\Sigma_\alpha^s(\mathbf{q}, i\omega) = -\frac{1}{2N_s} \sum_{\mathbf{k}, \alpha_1, \alpha_2 = \pm 1} \frac{|V_s^{(3)}(\mathbf{k}, \bar{\mathbf{k}} + \bar{\mathbf{q}}; \mathbf{q}; \alpha_1, \alpha_2, \alpha)|^2}{i\omega + \omega_{\mathbf{k}, \alpha_1} + \omega_{\mathbf{q} + \bar{\mathbf{k}}, \alpha_2}}, \quad (33)$$

respectively.

Finally, the diagrams that appear in the last line for both panels of Fig. 6 arise from the normal ordering of the quartic term $\mathcal{H}^{(4)}$ in Eq. (9). These contributions simply renormalize the quadratic Hamiltonian:

$$\mathcal{H}_{NO}^{(4)} = \sum_{\mathbf{q}, \alpha, \alpha'} [V_{\alpha\alpha'}^{(4,N)} \beta_{\mathbf{q}, \alpha}^\dagger \beta_{\mathbf{q}, \alpha'} + (V_{\alpha\alpha'}^{(4,A)} \beta_{-\mathbf{q}, \alpha} \beta_{\mathbf{q}, \alpha'} + h.c.)], \quad (34)$$

where $V_{\alpha\alpha'}^{(4,N)} (V_{\alpha\alpha'}^{(4,A)})$ represents the normal (anomalous) contributions. Since $\mathcal{H}_{NO}^{(4)}$ is of order $M^0$, only the diagonal normal contribution arising from the normal vertex $V_{\alpha\alpha'}^{(4,N)} \delta_{\alpha,\alpha'}$ gives a relative correction of order $1/M$ to the bare single-particle energy given in Eq. (21) (the anomalous terms in Eq. (34) give a relative correction contribution order $1/M^2$). The derivation of $V_{\alpha\alpha}^{(4,N)}$ is included in Note 7 of the Supplementary Information.

We note the parallel between the decay, sink, and quartic diagrams that give the $1/M$-correction to the single-particle self-energy and the ones that appear in the $1/S$-expansion of the standard SU(2) spin-wave theory of non-collinear Heisenberg magnets[11]. The main difference is that the SU(3) theory includes an extra bosonic flavor that enables more symmetry-allowed decay channels. In addition, the cubic–linear diagram exists even in absence of magnetic field because the magnitude of the ordered magnetic moment can be renormalized by changing the variational parameter θ. These diagrams, shown in the third line of Fig. 6a and the fourth line of Fig. 6b, are obtained by contracting one of the legs of the decay vertex with the cubic–linear vertex shown in Fig. 5. By using the Feynman rules, the cubic–linear diagrams are calculated as

$$\Sigma_\alpha^{cl}(\mathbf{q}) = -\frac{1}{\omega_{\mathbf{0},-1}} ([V_d^{(3)}(\mathbf{0}, \bar{\mathbf{q}}, \mathbf{q}; \alpha, -1, \alpha)]^* V_{L,\alpha} + h.c.). \quad (35)$$

By applying the analytic continuation $\omega \pm i\delta^+ \rightarrow i\omega$ and adopting the so-called on-shell approximation $\omega = \omega_\mathbf{q}$ for Eq. (32) and Eq. (33), the renormalized pole of the dressed propagator $\mathcal{G}$ is calculated as $\widetilde{\omega}_{\mathbf{q},\alpha} - i\widetilde{\Gamma}_{\mathbf{q},\alpha} = \omega_{\mathbf{q},\alpha} + V_{\mathbf{q},\alpha\alpha}^{(4,N)} + \Sigma_\alpha^{cl}(\mathbf{q}) + \Sigma_\alpha^s(\mathbf{q}, \omega_{\mathbf{q},\alpha}) + \Sigma_\alpha^d(\mathbf{q}, \omega_{\mathbf{q},\alpha})$, where the imaginary part of the pole $\widetilde{\Gamma}_{\mathbf{k},\alpha}$ arises from the decay term $\Sigma_\alpha^d$, that accounts for the observed broadening of the longitudinal mode in most regions of the BZ (see Fig. 3c, f) (the calculations are summarized in Note 9 of the Supplementary Information and ref. [52]). Moreover, the shift in the real part of the pole implies a corresponding renormalization in the model parameters. By fitting the neutron-scattering data with the renormalized dispersion peaks $\widetilde{\omega}_{\mathbf{q},\alpha}$ at the ZC, we obtain the set of optimal Hamiltonian parameters listed as set $\mathcal{B}$ in Table 1 and discussed further below.

## Discussion

**Comparison between experiment and theory.** To understand the spin excitation spectrum of $Ba_2FeSi_2O_7$ and demonstrate the importance of using the one-loop corrections, we start the comparison between experiment and theory with the GLSWT (i.e. without one-loop corrections). Figure 3b and e shows contour plots of $I(\mathbf{Q}, \omega)$ (Eq. (22)) calculated with the GLSWT along the $[H, 0, 0.5]$, and $[H, H, 0.5]$ direction, respectively. The Hamiltonian parameters (see set $\mathcal{A}$ in Table 1) are extracted by fitting the measured positions of the quasi-particle peaks (Gaussian-fitted peak centers of the experimental data) at the ZC. The GLSWT reproduces the dispersion of the observed two transverse modes $T_1$ and $T_2$ along the $[H, 0, 0.5]$ and $[H, H, 0.5]$ directions (Fig. 3b, e). Noticeably, the calculated longitudinal mode closely reproduces the experimental dispersion of the "$L$"-mode, which demonstrates that the SU(3) spin-wave theory describes the quasi-particles in $Ba_2FeSi_2O_7$.

Notably, the GLSWT does not reproduce the broadening and renormalization of the longitudinal modes observed in the inelastic neutron-scattering data. This is because the effect arises from the decay of a longitudinal mode into two transverse modes that is induced by the cubic term $\mathcal{H}^{(3)}$ of the expansion (Eq. (9)). To capture this effect, the $1/M$-correction from the one-loop

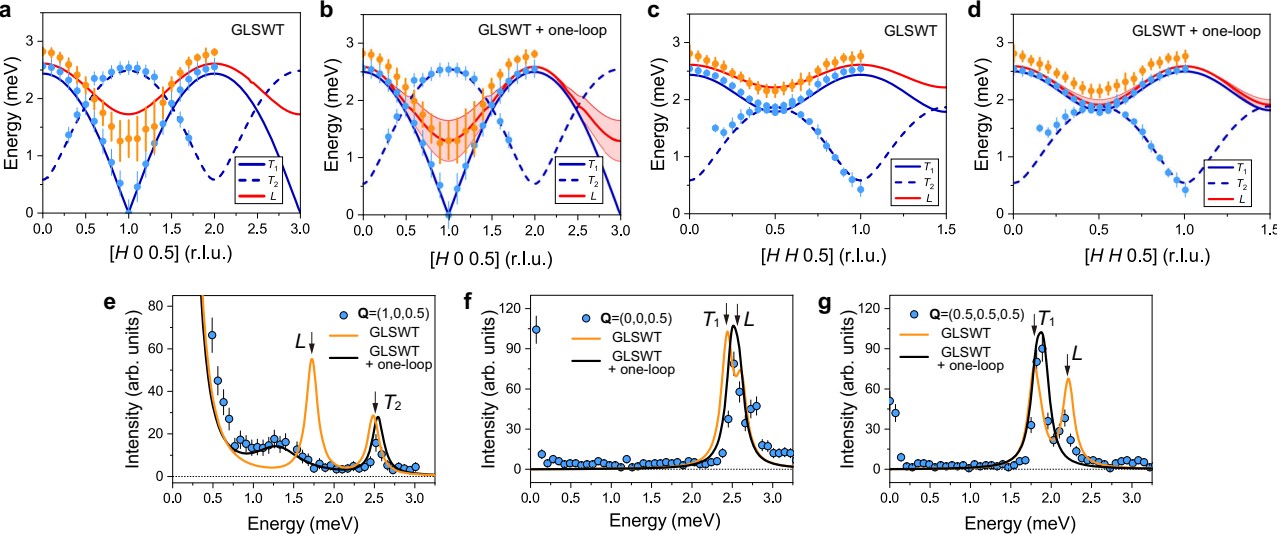

**Fig. 7 Comparison between measured and calculated spectrum.** Comparison of the measured and calculated dispersion along the [H, 0, 0.5] (**a**, **b**) and [H, H, 0.5] (**c**, **d**) directions. In all panels of this figure, the theoretical results are obtained for the parameter set $\mathcal{B}$ in Table 1. **a–d** Blue and orange filled circles indicate the measured transverse and longitudinal modes, obtained from the Gaussian fitting to the data shown in Fig. 4a. Dots and error bars indicate peak centers and full width at half maxima (FWHM) of the observed modes, respectively. Lines indicate the calculated dispersions obtained from the GLSWT and GLSWT+one-loop corrections. The red-shaded region in **b** and **d** depict the decay (line broadening) of the longitudinal mode given by the one-loop corrections. **e–g** Comparison between the measured (blue dots) and calculated (orange and black lines) INS intensities at three high-symmetric **Q**-points at (1, 0, 0.5), (0, 0, 0.5), and (0.5, 0.5, 0.5). All the experimental data were measured using CTAX with fixed $E_f = 3$ meV. For GLSWT, two transverse and longitudinal modes are denoted with $T_1$, $T_2$, and $L$.

expansion (see "Nonlinear correction" section) must be included. The GLSWT+one-loop correction can then describe the broadened spectrum of the longitudinal mode. The new Hamiltonian parameters, which are determined via the same procedure that is described above (see set $\mathcal{B}$ in Table 1), allow us to reproduce the observed spectrum (see Fig. 3c, f).

A more in-depth comparison between theory and experiment is shown in Fig. 7a and b. These figures show the quasi-particle dispersions along the [H, 0, 0.5] direction calculated with the GLSWT and GLSWT plus one-loop corrections compared to the measured dispersion. Near the ZC, $\mathbf{Q_m} = (1, 0, 0.5)$ the energy of longitudinal mode obtained from the GLSWT is noticeably higher than the peak center of the measured "$L$"-mode (orange dots). The discrepancy in the dispersion is resolved by introducing the one-loop corrections. The real part of the self-energy renormalizes the energy of the longitudinal mode, leading to a better agreement with the observed peak positions near the ZC. At the same time, the imaginary part of the self-energy obtained from the decay diagrams, $\Sigma_\alpha^d$, leads to an intrinsic line broadening of the longitudinal mode that is missing in the GLSWT. In Fig. 7b and d, the lower (upper) boundary of the red-shaded region is given by $\widetilde{\omega}_{\mathbf{k},-1}(\mp)\widetilde{\Gamma}_{\mathbf{k},-1}$, representing theoretical line broadening of the longitudinal mode that is compared against the experimental FWHM (orange error bars). In particular, the above-mentioned effects are most striking at $\mathbf{Q_m} = (1, 0, 0.5)$, therefore we present a comparison of the intensity line-cut at this momentum transfer in Fig. 7e. It is interesting to note that the energy shift of the transverse mode is also captured by the one-loop corrections.

After verifying that the one-loop corrections can simultaneously capture the broadening of the longitudinal mode and the energy shift of both the transverse and the longitudinal modes at the magnetic ZC, it is natural to ask if this also holds true far away from the ZC. Figure 7f, g is the intensity cuts for two representative points on the ZB. At a first glance, the peak centers of both modes are reasonably reproduced by the one-loop

corrections. A more detailed analysis reveals that the experimental FWHM of both peaks is equal to the instrumental resolution. However, as illustrated in Fig. 8a, since the longitudinal modes are still inside the two-magnon continuum, the one-loop correction predicts an intrinsic broadening (black curves) in Fig. 7f, g.

To understand the origin of this discrepancy, we trace back the decay channel of the longitudinal mode on the zone boundaries. The two-magnon continuum at the zone edge starts at an energy equal to the sum of the single-magnon energies at the zone center and the zone boundary. Due to the U(1) symmetry of the effective Hamiltonian, the magnons are gapless at the zone center, implying that the onset of the two-magnon continuum coincides with the single-magnon branch (see Fig. 8). In absence of U(1) symmetry, the magnon modes become gapped and the longitudinal mode does not need to lie inside the two-magnon continuum for arbitrary values of the wave vector (see Fig. 8b). A small magnon gap pushes the onset of the two-magnon continuum to be above the energy of the longitudinal mode at the zone boundaries. This modification of the two-magnon spectrum precludes the decay of the longitudinal mode near the zone boundary and explains the experimental observation. We then conjecture that the single-magnon dispersion is indeed gapped.

Unfortunately, it is difficult to extract the size of this gap from our INS data because of the large quasi-elastic scattering. Nevertheless, the analysis presented in Note 2 of the Supplementary Information indicates that our data are indeed consistent with a finite spin gap. We note that the gap can be captured by working with the original spin $S = 2$ Hamiltonian (Eq. (1)). The tetragonal symmetry allows for a single-ion anisotropy term of the form $\mathcal{H}_A = A\sum_i[(S_i^x)^4 + (S_i^y)^4]$, which breaks the global U(1) symmetry, generating a finite gap for the transverse mode. However, when we project the original $S = 2$ Hamiltonian onto the low-energy space to obtain the effective spin $S = 1$ Hamiltonian (Eq. (2)), the term $\mathcal{H}_A$ simply renormalizes the single-ion anisotropy, implying that the low-energy model

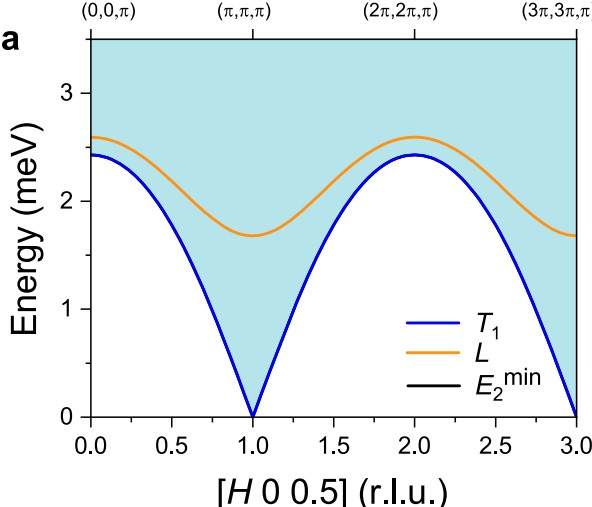

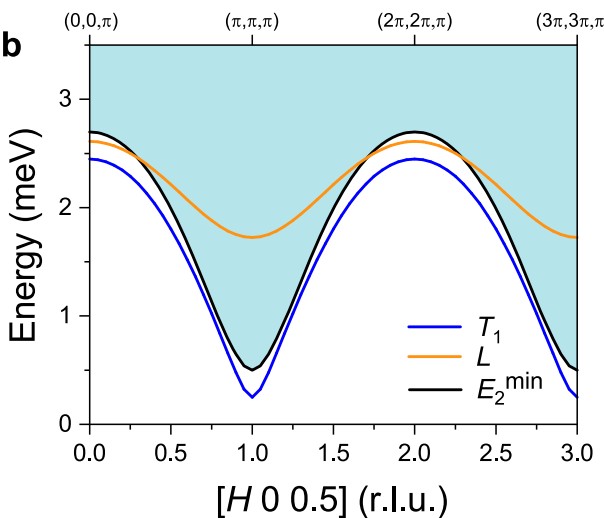

**Fig. 8 Kinematic constraints for the decay of the longitudinal mode.** The blue (orange) curve shows the calculated transverse (longitudinal) band dispersions along [H, 0, 0.5] with the GLSWT (using parameters set $\mathcal{B}$ in Table 1). The light blue-shaded areas indicate the two-transverse mode continuum, whose lower edge is indicated with a black solid line ($E_2^{min}$). **a** Results of the effective $S = 1$ model. **b** Same as **a** but for a gapped branch of transverse modes (an ad hoc gap has been added to Eq. (21)).

the Goldstone mode (see Note 8 in the Supplementary Information), although the individual contributions from the diagrams shown in Fig. 5 diverge as $1/q$ in the long-wavelength limit. We note that previous attempts of computing the decay of the longitudinal mode[34] have not accounted for the renormalization of the single-particle dispersion arising from the $1/M$-correction to the real part of the self-energy. This correction leads to a significant change in the extracted ratio $\alpha = \widetilde{J}/\widetilde{D}$ of $Ba_2FeSi_2O_7$, cf. $\alpha_{GLSWT} = 0.152$, and $\alpha_{GLSWT+one-loop} = 0.187$. This change is a direct consequence of the substantial renormalization of the energy $\omega_L(\mathbf{Q_m})$ of the longitudinal mode at the ZC. In fact, an accurate calculation that goes beyond the one-loop approximation estimates that the critical $\alpha_c$ required to close the gap $\omega_L(\mathbf{Q_m})$ for $\widetilde{J}' = 0.1\widetilde{J}$, and $\widetilde{\Delta} = \widetilde{\Delta}' = 1/3$ is around 0.158. In other words, the Hamiltonian parameters extracted from fitting the experiment with the GLSWT place $Ba_2FeSi_2O_7$ on the quantum paramagnetic side of the phase diagram shown in Fig. 1, which obviously contradicts the experimental evidence. In contrast, the set of parameters obtained from the GLSWT+one-loop correction ($\alpha_{GLSWT+one-loop}$) place the material at the magnetically ordered phase of the exact phase diagram. Furthermore, the calculated ordered moment is very close to the measured value 2.95 $\mu_B$ (see Note 12 of the Supplementary Information for discussion of the reduction of the ordered moment). In general, nonlinear corrections become increasingly important upon approaching the QCP and logarithmic corrections due to multi-loop vertex renormalizations become relevant very close to this point[28,31,53,54]. The fact that one-loop correction is enough to reproduce the spectrum of $Ba_2FeSi_2O_7$ indicates that this material is still far enough from that critical regime.

In summary, $Ba_2FeSi_2O_7$ provides a natural realization of a quasi-2D easy-plane antiferromagnet in the proximity of the QCP that signals the transition into the QPM phase. Previous examples of low-dimensional easy-plane quantum magnets in the proximity of this QCP were typically located on the quantum paramagnetic side of the quantum phase transition[17,20,21,23]. $Ba_2FeSi_2O_7$ then allows us to explain the strong decay and renormalization effects of the low-energy transverse and longitudinal modes of the AFM state. Furthermore, the distance to the O(2) QCP could be in principle controlled by chemical substitution, while the application of an in-plane magnetic field, that gaps out the transverse modes, can be used to control the decay rate of the longitudinal mode.

Here, we have used the INS data of $Ba_2FeSi_2O_7$ as a platform to test a loop expansion based on an SU(3) spin-wave theory[17,18,20,55], that captures the longitudinal and the transverse modes at the linear level. This loop expansion, which generalizes the well-known $1/S$-expansion of the SU(2) spin-wave theory, allows us to reproduce the measured width and renormalization of the longitudinal and transverse modes near the zone center by just including a one-loop correction. Small discrepancies near the zone boundary are attributed to limitations of the effective low-energy $S = 1$ model that we adopted for this work.

The loop expansion that we have described in this manuscript provides a general scheme for treating quantum magnets with more than one type of low-energy mode. In general, quantum magnets that exhibit low-energy modes with $N - 1$ different "flavors" can be treated semi-classically using an SU($N$) spin-wave theory. The parameter of the semi-classical expansion is the number of loops in the Feynman diagrams that contribute to the single-particle propagator.

acquires an "emergent" U(1) symmetry that is absent in the original high-energy model. Lastly, we note that the energies of the longitudinal mode on the zone boundaries after the one-loop corrections are slightly lower than the measured values. This level of discrepancy can be attributed to the missing second-order corrections $O\left(\frac{J^2}{3D}\right)$ to the low-energy model (2) or to missing terms in the original Hamiltonian (1). A simple analysis shows that a second nearest-neighbor AFM interaction with $\widetilde{J}_2 \sim 0.2\widetilde{J}$ can account for this discrepancy. For simplicity, $\widetilde{J}_2$ is not included in our calculation. Except for the discrepancy near the zone boundaries, the effective $S = 1$ model with one-loop corrections successfully captures most features of the INS data inside the BZ.

Finally, we emphasize that the loop expansion preserves the Goldstone mode that results from the spontaneous breaking of the emergent U(1) symmetry group of $\widetilde{\mathcal{H}}_{eff}$. More specifically, the $\mathcal{O}(M^0)$ correction to the real part of the self-energy vanishes for

## Methods

**Sample preparation.** A single crystal of $Ba_2FeSi_2O_7$ was grown using an optical floating zone melting method[44]. Polycrystalline $Ba_2FeSi_2O_7$ feed-rods were

prepared using the solid-state reaction method. The stoichiometric powders of $BaCO_3$ and $Fe_2O_3$, and $SiO_2$ were mixed, ground, pelletized, and sintered with intermediate heating in a reduced gas atmosphere. $Ba_2FeSi_2O_7$ single crystal was grown using a floating zone furnace in the same gas environment.

**Inelastic neutron-scattering measurement**. Inelastic neutron-scattering measurements were performed using the cold neutron triple-axis spectrometer (CTAX) at the High Flux Isotope Reactor (HFIR) and the hybrid spectrometer (HYSPEC) at the Spallation Neutron Source (SNS) at Oak Ridge National Laboratory[45]. A 2.15-g single crystal was aligned with the $(H, H, L)$ and $(H, 0, L)$ in the horizontal scattering plane for CTAX and HYSPEC experiments. A liquid helium cryostat was used to control temperature. At CTAX, the initial neutron energy was selected using a PG (002) monochromator, and the final neutron energy was fixed to $E_f = 3.0$ meV by a PG (002) analyzer. The horizontal collimation was guide-open-$40'-120'$, which provides an energy resolution with a full width half maximum (FWHM) $= 0.1$ and 0.18 meV for $\Delta E = 0$ and 2.5 meV, respectively. For the HYSPEC experiment, $E_i = 9$ meV and a Fermi chopper frequency of 300 Hz were used, which provides an energy resolution of FWHM $= 0.28$ meV and 0.19 meV at $\Delta E = 0$ and 2.5 meV, respectively. Measurements were performed at $T = 1.6$ K and 90 K by rotating the sample from $-50$ to 170° with 1° steps. Data were symmetrized over positive and negative $H$ and integrated over $K = [-0.1, 0.1]$ and $L = [0.4, 0.6]$. In Fig. 3a, there appears to be quasi-elastic scattering below 0.5 meV in low **Q**-region. This scattering arises from the incompletely blocked direct beam due to the oscillating collimator. All of the datasets were reduced and analyzed using MANTID[56] and DAVE[57].

## Data availability
The datasets generated during and/or analyzed during the current study are available from the corresponding authors on reasonable request.

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

## Acknowledgements

We thank Shang-Shun Zhang, Jie Xing, and Andrew F. May for useful discussions and Choongjae Won for helping with sample growth. This work was supported by the U.S. Department of Energy, Office of Science, Basic Energy Sciences, Materials Science, and Engineering Division. This research used resources at the High Flux Isotope Reactor and Spallation Neutron Source, DOE Office of Science User Facilities operated by the Oak Ridge National Laboratory (ORNL). Access to MACS was provided by the Center for High Resolution Neutron Scattering, a partnership between the National Institute of Standards and Technology and the National Science Foundation under Agreement No. DMR-1508249. The work at Max Planck POSTECH/Korea Research Initiative was supported by Nano Scale Optomaterials and Complex Phase Materials (2016K1A4A4A01922028) and Grant No. 2020M3H4A2084418, through the National Research Foundation (NRF) funded by MSIP of Korea. The work at Rutgers University was supported by the DOE under Grant No. DOE: DE-FG02-07ER46382. This manuscript has been authored by UT-Battelle, LLC under Contract No. DE-AC05-00OR22725 with the U.S. Department of Energy. The United States Government retains and the publisher, by accepting the article for publication, acknowledges that the United States Government retains a non-exclusive, paid-up, irrevocable, worldwide license to publish or reproduce the published form of this manuscript, or allow others to do so, for United States Government purposes. The Department of Energy will provide public access to these results of federally sponsored research in accordance with the DOE Public Access Plan (http://energy.gov/downloads/doe-public-access-plan).

## Author contributions

S.H.D. and A.D.C. conceived the project, which was supervised by C.D.B. and A.D.C. T.H.J., S.W.C., and J.-H.P. provided single crystals. T.H.J. measured physical properties. S.H.D., T.J.W., T.H., V.O.G., J.A.R., and A.D.C. performed INS experiments. S.H.D., T.J.W., and A.D.C. analyzed the neutron data. H.Z. and C.D.B. constructed theoretical model and calculations. S.-H.D., H.Z., C.D.B., and A.D.C. wrote the manuscript with input from all authors.

## Competing interests

The authors declare no competing interests.
