## [Peer Review File · Nature Communications]

Reviewers' Comments:

Reviewer #1:

Remarks to the Author:

Ba₂FeSi₂O₇ effectively features spin-1 moments coupled on an antiferromagnetic square lattice. High quality neutron spectroscopy measurements have been performed in the ordered state. The experimental results are simply gorgeous, with several modes clearly visible to the eye across the Brillouin zone in color QE-maps and a sharp enough resolution to clearly see broadening effects. A new spin wave theory was developed to generalize 1/S expansion to SU(3) symmetry. I'm an experimentalist and therefore not the best person to fully assess the details of the theory, but from my standpoint it appears valid and likely to have a pretty significant impact by proving useful in further studies of other S=1 quantum magnets.

Neutron powder diffraction data find an ordered moment that is only 74% of the expected moment for S=2 spins. It is said that this supports the effective S=1 picture. Can this be better explained? 74% isn't *that* small so it doesn't seem obvious to me that this fact strongly points to effective S=1 spins..

One problem is that the GLSWT+one loop correction calculations provide an obviously worse fit to the data near the zone boundary (and across the [H H 0.5] scans) than the simpler GLSWT model. The authors try to explain this discrepancy by pointing out that a fairly small spin wave gap, small enough to still be consistent with the data, would push the boundary of the two-mode continuum slightly above the zone-boundary edge of the longitudinal mode and thus blocking the broadening. I think that this argument does a reasonable job of explaining the width of the longitudinal mode near the zone boundary, but it is still the case that the simple GLSWT model does a demonstrably better job than the GLSWT+one loop correction model at predicting the position of the longitudinal mode near the zone boundary. Would blocking the two-mode decay push the calculated peak position up to match the data? Even better: Is it possible to perform a full GLSWT+one loop correction calculation with a small ad hoc spin wave gap added in? If additional curves could be added to Figures 7D and 7G showing calculations for the GLSWT+one loop model with a small gap that match the data reasonably well it would significantly strengthen the author's case.

Reviewer #2:

Remarks to the Author:

The authors provide a combined experimental and theoretical analysis of the low-lying excitations of a quantum antiferromagnet, Ba₂FeSi₂O₇. The material consists of weakly-coupled layers of S=2 spins, whose ordering into an XY antiferromagnet gives rise to collective modes - a gapless transverse magnon, a gapped transverse magnon, and a gapped longitudinal mode. The dispersion relations of these modes, as well as their lifetimes, are measured through inelastic neutron scattering. These measurements are complemented by a theoretical analysis based on an effective model of S=1 spins, which are analyzed through a one-loop expansion. In particular, the one-loop expansion is shown to give rise to important renormalization of the longitudinal mode energy and of its lifetime.

This is a solid contribution to the study of collective modes in antiferromagnets. The system studied is interesting and the questions being addressed are important. The theory presented is clear and the underlying assumptions are made explicit. The close collaboration between theory and experiment adds great value to the paper, as it allows for quantitative analysis of renormalization due to quantum fluctuations in this system.

However, prior to making a definitive recommendation regarding publication, I would like clarification of the following point: the longitudinal mode is known to become sharp close enough to the quantum critical point (QCP) (assuming 3+1 dimensions, due to the interlayer coupling which is relevant in the ordered phase). This occurs because interactions are marginally irrelevant at the QCP. In the one-loop expansion, which is strictly speaking only justified when M is large, the renormalization of the interaction strength which drives it to zero is not taken into account. Given that the system under study seems to be quite close to the QCP, is one justified in neglecting this

effect?

Reviewer #3:

Remarks to the Author:

After agreeing to referee the paper based on the abstract, which promises much experimental work by neutron scattering and a bit of theory to explain it, I was surprised to find that this is predominantly a theory paper. The paper mainly reports, in some detail, the development of a generalized spin wave theory. An experimental case that demonstrates the applicability of this theory is also reported. I don't feel I can comment on the correctness of this theory and I do feel that the abstract should reflect the balance and priorities of the paper!

Recent years have seen a realization that quantum magnetic effects occur in all manner of spin systems beyond those based on Cu^{2+} ($S=1/2$) and Ni^{2+} ($S=1$) in low-dimensional geometries. There has been much interest in $J_{\text{eff}}=1/2$ systems, and now, somewhat reminiscent, is gathering interest in systems with more complicated single-ion ground states that may well have interesting collective phenomena not captured by linear spin wave theory. In this sense, this theory (and other related applications) appear to be important. It is couched in the language of modern theory, but is nonetheless a pragmatic theory of real materials, developed with the microscopic physics of Fe^{2+} in $\text{Ba}_2\text{FeSi}_2\text{O}_7$ in view. I think it is valuable that sophisticated theories continue to be developed (and visibly reported), which can help to understand all manner of materials that land outside well known model classes (like the canonical quantum magnets mentioned above or those materials that can be entirely handled with conventional spin wave theory). In particular, as modern experimental techniques, in this case inelastic neutron scattering, reveal ever greater degrees of detail.

The experimental work is not the main focus of the work - it appears to be solid. I would address the following points:

On page 5, they describe the type of fluctuations in the T1 and T2 modes. These would have a distinct signature in a polarized neutron experiment that isn't mentioned, and the description of HYSPEC suggests it was used in unpolarized mode. Is the attribution of the fluctuations then a physical argument, an experimental observation, or extracted later from the theory but described here?

In the first paragraphs of the discussion I would use the same label/acronym in the text (GLSWT) as on the figure (currently GLSW).

The magnon gap should be revealed in a heat capacity measurement and if they have easy access to the relevant equipment, this might even be done quite quickly, on the timescale of resubmitting the manuscript. I would suggest that they could first evaluate the difference between the heat capacity of their gapless and gapped models and see if a regime with easily distinguishable power law forms can be accessed.

I am surprised to see that the instrumental resolution was modeled using a Lorentzian function, whereas a Gaussian is far more normal. What is the reason for this choice? Moreover, since some of the conclusions depend on accurate understanding of the resolution of the spectrometers used, I think the supplement is missing a section that describes how these parameters are arrived at and describes how they are incorporated into the comparison of experimental data and theoretical calculation. I appreciate that this will be quite standard stuff for a cold neutron triple axis spectrometer, but I don't think the same can be said for HYSPEC.

Much comparison of the theory and data rests on color maps as in Fig. 3. I would encourage them to explore some more quantitative assessments of the quality of the model. The supplementary information of Ref. 18 could provide examples - numerical/statistical quantification of the goodness of fit, and plots of calculated vs observed intensity (ideally actually the point-density in such a plot for data like this with many points).

The fitting process should be described, and the parameters should have some uncertainty

estimate.

Well known as the $1/S$ expansion and 1-loop correction may be, I think when they are introduced they should have definitive references for non-theoreticians who wish to understand them better.

Supplementary note 2 should refer to Fig. 2, not Fig. 1.

REVIEWER COMMENTS

Reviewer #1 (Remarks to the Author):

Ba₂FeSi₂O₇ effectively features spin-1 moments coupled on an antiferromagnetic square lattice. High quality neutron spectroscopy measurements have been performed in the ordered state. The experimental results are simply gorgeous, with several modes clearly visible to the eye across the Brillouin zone in color QE-maps and a sharp enough resolution to clearly see broadening effects. A new spin wave theory was developed to generalize 1/S expansion to SU(3) symmetry. I'm an experimentalist and therefore not the best person to fully assess the details of the theory, but from my standpoint it appears valid and likely to have a pretty significant impact by proving useful in further studies of other S=1 quantum magnets.

→ We thank the reviewer for recognizing the importance of the work.

Neutron powder diffraction data find an ordered moment that is only 74% of the expected moment for S=2 spins. It is said that this supports the effective S=1 picture. Can this be better explained? 74% isn't that small so it doesn't seem obvious to me that this fact strongly points to effective S=1 spins.

→ We thank the referee for pointing this out. We added a new note in the supplementary material (see Note 11), where we provide a detailed discussion of the calculated ordered moment and the comparison with the measured value. The short explanation is that the projection of the $S^{x,y}$ components of the spin 2 operator is $\sqrt{3} s^{xy}$. The $\sqrt{3}$ factor explains why the ordered moment obtained with the effective $S = 1$ Hamiltonian can be bigger than half of the full $S = 2$ moment. Also, we have revised the statement in the new version to avoid possible misunderstanding (Main text line 130).

One problem is that the GLSWT+one loop correction calculations provide an obviously worse fit to the data near the zone boundary (and across the [H H 0.5] scans) than the simpler GLSWT model. The authors try to explain this discrepancy by pointing out that a fairly small spin wave gap, small enough to still be consistent with the data, would push the boundary of the two-mode continuum slightly above the zone-boundary edge of the longitudinal mode and thus blocking the broadening. I think that this argument does a reasonable job of explaining the width of the longitudinal mode near the zone boundary, but it is still the case that the simple GLSWT model does a demonstrably better job than the GLSWT+one loop correction model at predicting the position of the longitudinal mode near the zone boundary. Would blocking the two-mode decay push the calculated peak position up to match the data? Even better: Is it possible to perform a full GLSWT+one loop correction calculation with a small ad hoc spin wave gap added in? If additional curves could be added to Figures 7D and 7G showing calculations for the GLSWT+one loop model with a small gap that match the data reasonably well it would significantly strengthen the author's case.

→ We thank the referee for bringing this up. This is an excellent point. Following the suggestion of the referee, we computed the effect of the small gap (~0.2meV) on the real part of the self-energy at the zone boundary. While it is certainly true that this correction reduces the difference between the calculated and measured energies of the longitudinal mode, the magnitude of the effect (the relative shift of the longitudinal mode is 1%) is still not enough to eliminate the discrepancy. It is important to mention that the effective $S = 1$ model is expected to reproduce the low-energy spectrum of the original $S = 2$ model with an error of about 10%. The basic reason is that our low-energy $S = 1$ model does not include second-order corrections $O(J^2/3D)$ that generate effective spin interactions, which are not included in

Eq.(2) of the manuscript. We have added one sentence to the conclusions in order to clarify this important point.

Reviewer #2 (Remarks to the Author):

The authors provide a combined experimental and theoretical analysis of the low-lying excitations of a quantum antiferromagnet, Ba₂FeSi₂O₇. The material consists of weakly-coupled layers of S=2 spins, whose ordering into an XY antiferromagnet gives rise to collective modes - a gapless transverse magnon, a gapped transverse magnon, and a gapped longitudinal mode. The dispersion relations of these modes, as well as their lifetimes, are measured through inelastic neutron scattering. These measurements are complemented by a theoretical analysis based on an effective model of S=1 spins, which are analyzed through a one-loop expansion. In particular, the one-loop expansion is shown to give rise to important renormalization of the longitudinal mode energy and of its lifetime.

This is a solid contribution to the study of collective modes in antiferromagnets. The system studied is interesting and the questions being addressed are important. The theory presented is clear and the underlying assumptions are made explicit. The close collaboration between theory and experiment adds great value to the paper, as it allows for quantitative analysis of renormalization due to quantum fluctuations in this system.

→ We thank the reviewer for recognizing the importance of the work.

However, prior to making a definitive recommendation regarding publication, I would like clarification of the following point: the longitudinal mode is known to become sharp close enough to the quantum critical point (QCP) (assuming 3+1 dimensions, due to the interlayer coupling which is relevant in the ordered phase). This occurs because interactions are marginally irrelevant at the QCP. In the one-loop expansion, which is strictly speaking only justified when M is large, the renormalization of the interaction strength which drives it to zero is not taken into account. Given that the system under study seems to be quite close to the QCP, is one justified in neglecting this effect?

→ We thank the referee for bringing this important point to our attention. Indeed, since the dimension of the low-energy theory that describes the QCP coincides with the upper critical dimension, $D = 3 + 1$, multiloop vertex renormalizations produce logarithmic corrections, which are of course absent in the one-loop calculations. These corrections affect physical quantities such as the ratio between the decay rate $\tilde{\Gamma}_{q,-1}$ and the energy of the longitudinal mode, which is logarithmically suppressed at the QCP [1]. However, by their nature, logarithmic corrections become significant when the system is *extremely* close to the QCP, i.e., when the *gap* of the longitudinal mode Δ_L (in units of the exchange constant) and the staggered magnetization M_S are vanishingly small [1]. This is not the case of the system under consideration for which $\Delta_L \simeq 1.30$ meV and $M_S \simeq 2.95\mu_B$. We clarify this important point in the new version of the manuscript and we are also including references to papers that analyze the effect of the logarithmic corrections starting from an effective field theory (long-wavelength limit of the GSWT that is discussed in the manuscript) (Main text line 433).

We would also like to stress that vertex renormalizations that account for the above-mentioned logarithmic corrections involve a partial summation of diagrams with arbitrary number of loops [2]. This partial summation does not preserve the Goldstone mode because one is not including all the diagrams up to a given order in 1/M. Correspondingly, as suggested by the referee, the problem of computing the

decay and renormalization of the modes close enough to the QCP requires a non-perturbative treatment, which is beyond the scope of our manuscript.

[1] Y. Kulik and O. P. Sushkov Phys. Rev. B 84, 134418 (2011)

[2] H. D. Scammell and O. P. Sushkov, Phys. Rev. B 92, 220401(R) (2015)

Reviewer #3 (Remarks to the Author):

After agreeing to referee the paper based on the abstract, which promises much experimental work by neutron scattering and a bit of theory to explain it, I was surprised to find that this is predominantly a theory paper. The paper mainly reports, in some detail, the development of a generalized spin wave theory. An experimental case that demonstrates the applicability of this theory is also reported. I don't feel I can comment on the correctness of this theory and I do feel that the abstract should reflect the balance and priorities of the paper!

→ We are sorry that our abstract left the reviewer with this impression. We have attempted to adjust the abstract to better convey the contents of our paper. We do point out that there is considerable experimental work in the paper which we believe provides a data set of unprecedented quality in this area. Furthermore, we note that because of the possibility of direct comparison with detailed theory calculations, a more traditional paper where much time is spent on experimental aspects was not necessary—i.e. the theory allows deep insights to be extracted.

Recent years have seen a realization that quantum magnetic effects occur in all manner of spin systems beyond those based on Cu^{2+} ($S=1/2$) and Ni^{2+} ($S=1$) in low-dimensional geometries. There has been much interest in $J_{\text{eff}}=1/2$ systems, and now, somewhat reminiscent, is gathering interest in systems with more complicated single-ion ground states that may well have interesting collective phenomena not captured by linear spin wave theory. In this sense, this theory (and other related applications) appear to be important. It is couched in the language of modern theory, but is nonetheless a pragmatic theory of real materials, developed with the microscopic physics of Fe^{2+} in $\text{Ba}_2\text{FeSi}_2\text{O}_7$ in view. I think it is valuable that sophisticated theories continue to be developed (and visibly reported), which can help to understand all manner of materials that land outside well-known model classes (like the canonical quantum magnets mentioned above or those materials that can be entirely handled with conventional spin wave theory). In particular, as modern experimental techniques, in this case inelastic neutron scattering, reveal ever greater degrees of detail.

→ We thank the reviewer for recognizing the importance of the work.

The experimental work is not the main focus of the work - it appears to be solid. I would address the following points:

On page 5, they describe the type of fluctuations in the T_1 and T_2 modes. These would have a distinct signature in a polarized neutron experiment that isn't mentioned, and the description of HYSPEC suggests it was used in unpolarized mode. Is the attribution of the fluctuations then a physical argument, an experimental observation, or extracted later from the theory but described here?

→ Thank you for pointing this out. The T_1 - and T_2 -modes are identified via a theoretical interpretation of the data. We have made this more explicit in the new version of the manuscript.

In the first paragraphs of the discussion, I would use the same label/acronym in the text (GLSWT) as on the figure (currently GLSW).

→ We thank the reviewer for noticing this problem. We have corrected this typo in the revised version of the manuscript.

The magnon gap should be revealed in a heat capacity measurement and if they have easy access to the relevant equipment, this might even be done quite quickly, on the timescale of resubmitting the manuscript. I would suggest that they could first evaluate the difference between the heat capacity of their gapless and gapped models and see if a regime with easily distinguishable power law forms can be accessed.

→ We thank the reviewer for the suggestions. Specific heat can indeed provide information about the spin gap. However, the low-temperature specific heat may be sensitive to extrinsic effects such as oxygen vacancies of the sample (particularly Fe^{2+} compound) or phonon subtraction, and these terms can become critical when the magnetic contribution is exponentially small. As an alternative approach, we have measured the angular field dependence of the magnetic susceptibility to check for anisotropy consistent with a single-ion anisotropy term stemming from the $S = 2$ Hamiltonian that can induce a spin gap. As expected, below the Neel temperature ($T_N=5.2$ K), the magnetic susceptibility shows a weak easy-axis anisotropy within the plane. A detailed description of these results has been added in Note 3 of Supplementary information.

We have also added the field-dependent inelastic neutron scattering results in Note 2 of Supplementary information. Since the field dependence of the spin gap is known, the zero-field gap can be obtained by extrapolating the finite field gap to zero field. Our field-dependent spectrum shows a clear offset, which confirms the existence of a zero-field gap of approximately 0.18 (2) meV.

I am surprised to see that the instrumental resolution was modeled using a Lorentzian function, whereas a Gaussian is far more normal. What is the reason for this choice? Moreover, since some of the conclusions depend on accurate understanding of the resolution of the spectrometers used, I think the supplement is missing a section that describes how these parameters are arrived at and describes how they are incorporated into the comparison of experimental data and theoretical calculation. I appreciate that this will be quite standard stuff for a cold neutron triple axis spectrometer, but I don't think the same can be said for HYSPEC.

→ We thank the referee for bringing these important points to our attention. We have used a Lorentzian broadening to describe the instrumental resolution because such broadening can be 'naturally' incorporated in the theoretical calculations by adding an imaginary part to the frequency ω . While it is true that a Gaussian broadening is more accurate to reproduce the instrumental resolution, the difference between both distributions (they have different tails) is not significant in describing the line shape of the longitudinal mode. Quantitative comparison of the broadening effects has been newly added in Note 9 of Supplementary information, which confirms that the Lorentzian broadened longitudinal mode essentially has the same peak profile as the Gaussian broadened one, for our instrumental resolution. Correspondingly, for the particular case under consideration, the choice of Gaussian vs. Lorentzian distribution makes no practical difference.

We are sorry for missing the part about how parameters were obtained systematically. We have added Note 10 of Supplementary information describing the fitting process and how the parameters were obtained.

Much comparison of the theory and data rests on color maps as in Fig. 3. I would encourage them to explore some more quantitative assessments of the quality of the model. The supplementary information

of Ref. 18 could provide examples - numerical/statistical quantification of the goodness of fit, and plots of calculated vs observed intensity (ideally actually the point-density in such a plot for data like this with many points). The fitting process should be described, and the parameters should have some uncertainty estimate.

→ We thank the reviewer for pointing this out. Since the 'non-linear' corrections (GLSWT+one-loop correction) are computationally expensive and the model under consideration is a low-energy effective model, we fit the experimental data near the zone center (low-energy part of the spectrum), and used the rest of the data (mode dispersion and intensity far from the zone center and staggered magnetization) to test (validate) the resulting model. Details of the fitting procedure and validation tests are given in Notes 11 and 12 of the new Supplementary information. We have also estimated the uncertainty of the Hamiltonian parameters and provided the corresponding error bars.

Well known as the $1/S$ expansion and 1-loop correction may be, I think when they are introduced they should have definitive references for non-theoreticians who wish to understand them better.

→ We thank the reviewer for the kind suggestion. We have added multiple references to textbooks, review articles, and seminal works on the $1/S$ expansion. We are also citing the book by Sydney Coleman, "*Aspects of Symmetry*", Cambridge (1985), when we introduce the loop expansion (see line 192 in the new manuscript).

Supplementary note 2 should refer to Fig. 2, not Fig. 1.

→ We thank again the reviewer for noticing this typo that has been corrected it in the revised version of the manuscript.

Reviewers' Comments:

Reviewer #1:

Remarks to the Author:

This paper reports a theoretical advance that generalizes a $1/S$ expansion to probe collective excitations in an effective spin-1 quantum magnet. This theory is used to analyze high quality neutron data on Ba₂FeSi₂O₇ and accurately model the position and broadening of the longitudinal mode.

This paper features nice experimental data and a theoretical framework that could be useful in studying other spin-1 antiferromagnets. The authors have responded fairly well to the questions raised in the first round of review.

Reviewer #2:

Remarks to the Author:

I am satisfied with the authors' response to my comments and to those of the other referees and I find that the modifications made to the manuscript have increased its clarity. Hence, I recommend publication.

Reviewer #3:

Remarks to the Author:

I thank the authors for addressing my comments and those of the other referees. I think that most points have been addressed and the paper could be accepted for publication.

The authors might like to consider these small points:

The title/caption of Supplementary Fig. 3 mentions the magnetic specific heat, which does not actually appear on the figure.

In Supplementary Fig. 2c and related remarks, the linear fit is for a strictly gapless mode while the parabola is the appropriate model for a gapped mode. Why not fit the linear dependence with strictly zero intercept and show that the parabola with gap is a better fit to the data (and report the χ^2 for the two fits as well)? At present the small gap from the linear fit, which has no physical meaning, confuses the point a bit, and the two seem to fit the data equally well.

The fact that the Lorentzian broadening reproduces the spectrum, given the form of the spectra in Supplementary Fig. 5 seems to suggest that the L mode is broadened well outside the instrumental resolution. This is also suggested by its fitted width being almost three times greater than the T mode in an energy window where I wouldn't expect the spectrometer resolution to change so drastically. If the broadening of the mode is intrinsic, this would also support a short lifetime due to the the proximity of the QCP, as mentioned in the caption of Fig. 1.

REVIEWERS' COMMENTS

Reviewer #1 (Remarks to the Author):

This paper reports a theoretical advance that generalizes a $1/S$ expansion to probe collective excitations in an effective spin-1 quantum magnet. This theory is used to analyze high quality neutron data on Ba₂FeSi₂O₇ and accurately model the position and broadening of the longitudinal mode. This paper features nice experimental data and a theoretical framework that could be useful in studying other spin-1 antiferromagnets. The authors have responded fairly well to the questions raised in the first round of review.

→ We thank the reviewer for recognizing the importance of the work and providing constructive criticism.

Reviewer #2 (Remarks to the Author):

I am satisfied with the authors' response to my comments and to those of the other referees and I find that the modifications made to the manuscript have increased its clarity. Hence, I recommend publication.

→ We thank the reviewer for the constructive comments.

Reviewer #3 (Remarks to the Author):

I thank the authors for addressing my comments and those of the other referees. I think that most points have been addressed and the paper could be accepted for publication.

The authors might like to consider these small points:

The title/caption of Supplementary Fig. 3 mentions the magnetic specific heat, which does not actually appear on the figure.

→ We thank the reviewer for noticing this problem. We have corrected title/caption of the figure in the revised version of the manuscript.

In Supplementary Fig. 2c and related remarks, the linear fit is for a strictly gapless mode while the parabola is the appropriate model for a gapped mode. Why not fit the linear dependence with strictly zero intercept and show that the parabola with gap is a better fit to the data (and report the χ^2 for the two fits as well)? At present the small gap from the linear fit, which has no physical meaning, confuses the point a bit, and the two seem to fit the data equally well.

→ We thank the reviewer for pointing this out. The fitting of the field dependent the T_1 -mode to the gapless linear dependence and gapped parabola function are now compared in the new Supplementary Note 2. The parabola function presents better fitting result (lower χ^2 value) than gapless linear dependence's, which indicates the presence of spin gap.

The fact that the Lorentzian broadening reproduces the spectrum, given the form of the spectra in Supplementary Fig. 5 seems to suggest that the L mode is broadened well outside the instrumental resolution. This is also suggested by its fitted width being almost three times greater than the T mode in an energy window where I wouldn't expect the spectrometer resolution to change so drastically. If the

broadening of the mode is intrinsic, this would also support a short lifetime due to the proximity of the QCP, as mentioned in the caption of Fig. 1.

→ We agree with this comment and thank the reviewer for pointing this out.